# MemVLT: Vision-Language Tracking with Adaptive Memory-based Prompts

**Xiaokun Feng**[1,2]    **Xuchen Li**[1,2]    **Shiyu Hu**[5]    **Dailing Zhang**[1,2]
**Meiqi Wu**[3]    **Jing Zhang**[2]    **Xiaotang Chen**[1,2,4]    **Kaiqi Huang**[1,2,4]

[1]School of Artificial Intelligence, University of Chinese Academy of Sciences
[2]Institute of Automation, Chinese Academy of Sciences
[3]School of Computer Science and Technology, University of Chinese Academy of Sciences
[4]Center for Excellence in Brain Science and Intelligence Technology, Chinese Academy of Sciences
[5]School of Physical and Mathematical Sciences, Nanyang Technological University
{fengxiaokun2022, lixuchen2024}@ia.ac.cn, shiyu.hu@ntu.edu.sg, zhangdailing2023@ia.ac.cn
wumeiqi18@mails.ucas.ac.cn, jing_zhang@ia.ac.cn, {xtchen,kaiqi.huang}@nlpr.ia.ac.cn

## Abstract

Vision-language tracking (VLT) enhances traditional visual object tracking by integrating language descriptions, requiring the tracker to flexibly understand complex and diverse text in addition to visual information. However, most existing vision-language trackers still overly rely on initial fixed multimodal prompts, which struggle to provide effective guidance for dynamically changing targets. Fortunately, the Complementary Learning Systems (CLS) theory suggests that the human memory system can dynamically store and utilize multimodal perceptual information, thereby adapting to new scenarios. Inspired by this, (**i**) we propose a **Mem**ory-based **V**ision-**L**anguage **T**racker (**MemVLT**). By incorporating memory modeling to adjust static prompts, our approach can provide adaptive prompts for tracking guidance. (**ii**) Specifically, the memory storage and memory interaction modules are designed in accordance with CLS theory. These modules facilitate the storage and flexible interaction between short-term and long-term memories, generating prompts that adapt to target variations. (**iii**) Finally, we conduct extensive experiments on mainstream VLT datasets (*e.g.*, MGIT, TNL2K, LaSOT and LaSOT$_{ext}$). Experimental results show that MemVLT achieves new state-of-the-art performance. Impressively, it achieves 69.4% AUC on the MGIT and 63.3% AUC on the TNL2K, improving the existing best result by 8.4% and 4.7%, respectively. The code and models will be released at: https://github.com/XiaokunFeng/MemVLT.

## 1 Introduction

The vision-language tracking (VLT) task [1] aims to locate a user-defined object in a video sequence using multimodal prompts, which comprise a template patch and a language description. As an extension of traditional visual single object tracking (SOT) task [2, 3, 4], VLT can harness the complementary advantages of multiple modalities. Therefore, vision-language trackers (VLTs) have the potential to achieve more promising tracking performance, which has recently attracted widespread attention [5, 6, 7, 8].

Similar to SOT, VLT still adopts the one-shot setting [9], providing prompts only at the initial moment. However, these fixed prompts struggle to provide continuous reference for targets in video sequences due to their inherent dynamic variability [10]. As shown in Fig. 1 (a), the initial prompts depict a gun placed on a table. However, when the target is subsequently picked up, there is a significant deviation in the target's state from given prompts. Regarding the target state in different frames, we

38th Conference on Neural Information Processing Systems (NeurIPS 2024).

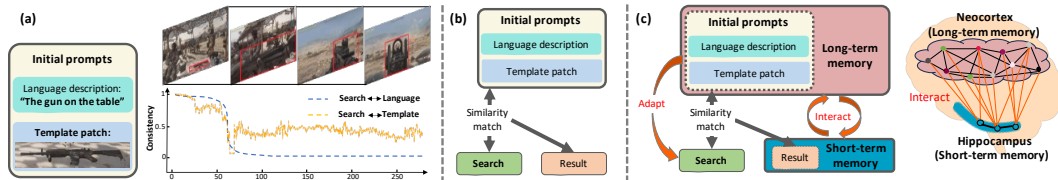

Figure 1: (a) Illustration of a video sequence. Given the initial prompts, we respectively plot consistency curves between prompts of two modalities and the subsequent searched target. (b) Framework of previous vision-language trackers (VLTs). They primarily obtain tracking results by matching the search image with the initial prompts based on similarity. (c) Framework of our proposed **MemVLT** (left) by modeling Complementary Learning Systems (CLS) Theory (right). MemVLT effectively models the storage and interaction of long-term and short-term memories, resulting in prompts that adapt to the search image.

quantitatively evaluate its consistency with the initial multimodal prompts (implementation details are introduced in Sec. B.1). It is evident that the consistency is poor most of the time, indicating that the initial prompts are ineffective as a reference.

Despite this, most existing VLTs [11, 12, 13, 14, 15, 16] overly rely on these static prompts. As illustrated in Fig. 1 (b), they identify the target in each frame that is most similar to the initial prompts as the tracking result. Despite achieving some success, the difficulty lies in the limited generalizability of static prompts to subsequent frames, thereby impacting tracking performance.

Unlike existing VLTs, humans can adaptively locate a target, even when the target appears in a form vastly different from its initial perception state [17]. Neuroscientists have long been interested in this ability and have conducted studies using the visual search task [18, 19], which is similar to the VLT. Numerous research findings reveal that human adaptive visual search capability can be attributed to the sophisticated memory mechanisms [20, 21].

The Complementary Learning Systems (CLS) theory [22], as a well-known memory model, has recently revealed the relationship between human generalization adaptability and memory [23]. Generally speaking, humans continuously adjust their memory by integrating perceived information to better adapt to the environment. As shown in Fig. 1 (c), the human brain achieves memory through two complementary systems: the hippocampus and the neocortex. The hippocampus plays a primary role in short-term memory, which is subsequently consolidated into long-term memory stored in the neocortex [24, 23]. The interaction between short-term and long-term memories enables humans to adapt to different environments [23, 8]. Given the advantages of this memory mechanism, a critical question arises: ***How can we incorporate it into the design of trackers?***

To achieve that, we propose a **Mem**ory-based **V**isual-**L**anguage **T**racker named **MemVLT**. As depicted in Fig. 1 (c), the core insight of MemVLT lies in efficiently adapting to the dynamic changes of the target by emulating the storage and interaction of memory information, thereby enabling effective tracking. Specifically, (**i**) we first develop a *memory storage module* to simulate the functioning of short-term and long-term memory systems. Inspired by the system consolidation process [23], we introduce an efficient long-term memory storage method named *section-top*. (**ii**) Drawing upon the stored memory, we incorporate a *memory interaction module* to emulate the interaction between short-term and long-term memories [25], generating adaptive visual and textual prompts. These adaptive prompts then guide subsequent tracking by integrating with the search feature. Through these modules, MemVLT facilitates adaptive tracking by utilizing memory information.

In summary, our contributions are as follows: (**i**) Inspired by the CLS theory, we introduce a novel tracker named MemVLT. Leveraging memory mechanism modeling, this approach facilitates the generation of adaptive prompts to effectively guide the tracking process. (**ii**) In the proposed MemVLT, we incorporate *memory storage* and *memory interaction* modules. They faithfully model the storage and flexible interaction processes between short-term and long-term memories in the human brain, yielding multimodal prompts that adapt to dynamically evolving targets. (**iii**) We evaluate the performance of MemVLT on the MGIT [5], TNL2K [26], LaSOT [27] and LaSOT$_{ext}$ [28] and achieve state-of-the-art tracking results. Notably, MemVLT achieves 69.4% AUC on MGIT and 63.3% AUC on TNL2K, improving the existing best result by 8.4% and 4.7%, respectively.

## 2 Related Work

### 2.1 Vision-Language Tracking

Vision-language tracking is an emerging multimodal task that aims to achieve tracking by utilizing a given language description and an initial template patch. Adhering to the principle of similarity-matching [9], most existing VLTs [11, 12, 13, 14, 15, 16] leverage the static given prompts as reference to identify the most similar target in the search frame. Among them, SNLT [11] introduces a universal language region proposal network, which improves tracking performance through a dynamic aggregation module. MMTrack [16] designs an effective tracking pipeline that treats the VLT as a token generation task. While these VLTs perform well in simple scenes, they overlook the dynamic nature of targets, making it challenging to track when targets undergo significant changes.

To address this limitation, certain VLTs attempt to utilize temporal information to obtain the dynamic reference. GTI [29] and AdaSwitcher [26] locate the target by integrating tracking and grounding results at each time step. Their reliance on pre-defined threshold can lead to error accumulation. JointNLT [14] incorporates temporal information in the form of a temporal query during the prediction stage. Since the length of the query tokens is much shorter than that of the static prompt tokens, it indicates that static prompts still dominate the tracking process. Differing from them, we depart from human memory mechanisms to utilize temporal information. Building upon initial static prompts, we inject dynamic features by incorporating memory information, which not only avoids the error accumulation but also generates prompts adapted to the dynamic changes of the target.

By treating temporal information as memory, some trackers [30, 31, 32] have incorporated memory mechanism modeling. Extensive previous studies confirm that memory modeling is a viable approach, which is the paradigm our work follows. Distinguishing our work from existing research, to our knowledge, MemVLT is the first to apply CLS theory to tracker design. Recent research in CLS theory [33, 23] underscores the importance of the interaction between long and short-term memories. Motivated by this insight, we develop the *memory interaction module* to generate adaptive prompts. In contrast, existing approaches, such as DecoupleTNL [32], focus solely on modeling long and short-term memories but overlook the interaction process. In addition to memory interaction, we propose the *section-top* memory storage method, which is inspired by the system consolidation process [23, 24] within the CLS theory. Compared to the *sliding window* storage method commonly used by existing trackers [14, 32], it demonstrates more effective tracking performance (see Sec. 4.3).

### 2.2 Prompt Learning

In the natural language processing field, prompt learning [34] refers to the automatic learning of instructions in the form of sentences, thereby enabling better task understanding. Considering the significant advantages of this approach in enhancing model adaptability, recent studies have extended it to vision-language tasks [35, 36]. CoOp [35] efficiently fine-tunes CLIP [37] for few-shot transfer by constructing language branch inputs using a set of learnable vectors. Bahng [36] performs prompt tuning on CLIP by prompting on the vision branch. In this work, we exploit the core idea of prompt learning by introducing learnable vectors to model the memory information. These vectors are used to adjust the initial static prompts, making them adaptive to the dynamic changes of the target.

Recently, several vision-only trackers [38, 39, 40] have attempted to model temporal information using learnable vectors. These vectors, also referred to as temporal queries, are utilized to capture global information at each moment and guide the subsequent tracking process. What sets us apart is that, (**i**) we focus on the vision-language multimodal scenario, not only modeling temporal queries but also emphasizing their adjustment to static prompts to comprehensively leverage temporal information. (**ii**) We design a novel multimodal query storage method called *section-top* to enhance the existing *sliding window* storage approach.

## 3 Methodology

### 3.1 Overview

The framework of MemVLT is depicted in Fig. 2 (a). Given two static prompts (text description and template patch) and the search image at a general time step ($t > 0$), the text and vision encoders first

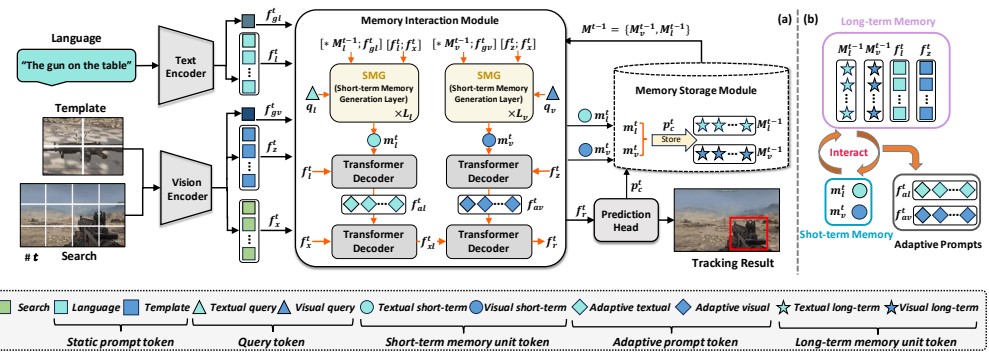

Figure 2: **(a) Framework of our proposed MemVLT.** Given a language description and a template patch as references, MemVLT tracks the target in search images at time $t$. The input is first encoded using *text* and *vision encoders*. Subsequently, the *memory interaction module* processes static prompt features based on stored memory, generating adaptive prompts. After incorporating these prompts, the search features are fed into the *prediction head* to obtain the tracking results. Additionally, the *memory storage module* utilizes processed data to represent and store memory information. **(b) Diagram of memory interaction.** To illustrate the process of memory interaction, we organize the memory information from the perspectives of long-term and short-term memories. Through the interaction between these memories, adaptive visual and textual prompts are obtained.

embed them into specific feature spaces, denoted as $f_l^t$, $f_z^t$, and $f_x^t$, respectively. Concurrently, we obtain the global visual and textual semantic representation tokens (*i.e.*, $f_{gv}^t$ and $f_{gl}^t$). Subsequently, these encoded features, along with the memory information $M^{t-1}$ obtained from the memory storage module (MSM), are fed into the memory interaction module (MIM). Through the mutual interaction of long-term and short-term memories, we obtain the latest visual and textual short-term memories (*i.e.*, $m_v^t$ and $m_l^t$), as well as the adaptive prompts (*i.e.*, $f_{av}^t$ and $f_{al}^t$). These adaptive prompts then undergo feature fusion with the search feature, yielding the target-related feature $f_r^t$. Next, $f_r^t$ is fed into the prediction head to obtain the tracking result and its corresponding confidence score $p_c^t$. Finally, $p_c^t$ and the short-term memory are used to update the MSM, providing memory information $M^t$ for tracking in the next time step $(t + 1)$. In the following sections, we will provide detailed introductions to each module and the dimensions of the above variables.

## 3.2 Input Encoder

**Vision Encoder.** For the visual input of the search image and template patch, we adopt the encoding paradigm of the one-stream network [41], implementing it with the HiViT [42, 39, 40]. Specifically, the template patch $z \in \mathbb{R}^{3 \times H_z \times W_z}$ and search image $x_t \in \mathbb{R}^{3 \times H_x \times W_x}$ are first projected into the feature space and flattened to produce token embeddings. Additionally, we introduce a [CLS] token[1] to capture global semantic feature [43]. This token is concatenated with the template-search tokens and then fed into transformer layers for feature modeling. Finally, we obtain the search feature $f_x^t \in \mathbb{R}^{N_x \times D}$, template feature $f_z^t \in \mathbb{R}^{N_z \times D}$, and global visual feature $f_{gv}^t \in \mathbb{R}^{1 \times D}$.

**Text Encoder.** Since being proposed, the BETR [43] series models have been widely used for text representation. Therefore, we utilize RoBERTa [44], a classic pre-trained model, as our text encoder. Specifically, for a given input sentence, we tokenize it into a sequence of text tokens. The token sequences then are fed into the RoBERTa to extract the text embedding feature $f_l^t \in \mathbb{R}^{N_l \times D}$. Additionally, we perform pooling on $f_l^t$ to obtain the sentence-level global feature $f_{gl}^t \in \mathbb{R}^{1 \times D}$.

## 3.3 Memory Interaction Module

Due to the constraints of initial static visual-textual prompts, the $f_l^t$ and $f_z^t$ derived from them face challenges in offering consistent references for dynamically changing targets [9]. To alleviate this limitation, MIM attempts to leverage the long-term memory $M^{t-1}$ stored in MSM to inject dynamic

---

[1]It is worth noting that the symbols CLS and [CLS] represent two distinct concepts in this paper. The former denotes the Complementary Learning Systems theory, while the latter represents the classification token [43].

feature reflecting target variations into the static prompts, thereby better guiding the tracking. As shown in Fig. 2 (a), under the influence of long-term memories (*i.e.*, $M_l^{t-1}$ and $M_v^{t-1}$), we first integrate with the current features to acquire the latest short-term memories (*i.e.*, $m_l^t$ and $m_v^t$). Then, through the interaction between the memory information $m^t$ and the initial prompts, we obtain adaptive prompts and fuse them with the search features $f_x^t$, yielding target-related features $f_r^t$.

Corresponding to the two modal prompts of given cues, we utilize both modalities to represent memory. Before explaining the specific construction of memory, we reiterate the short-term and long-term memory information that our model focuses on, as shown in Fig. 2 (b). For the long-term memory provided by MSM, denoted as $M^t = \left\{m^i\right\}_{i=1}^{L_m}$, it consists of $L_m$ short-term memory units $m^i = (m_v^i, m_l^i)$. Here, $m_v^i \in \mathbb{R}^{1 \times D}$ and $m_l^i \in \mathbb{R}^{1 \times D}$ represent the $i$-th short-term visual and textual memory, respectively. Additionally, we also consider the initial given prompts as a form of long-term memory due to their role throughout the tracking process.

### 3.3.1 The Acquisition of the Short-Term Memory

To begin with, we introduce the generation of short-term memory under the influence of long-term memory. Initially, we concatenate the long-term visual memory $\left\{m_v^i\right\}_{i=1}^{L_m}$ and textual memory $\left\{m_l^i\right\}_{i=1}^{L_m}$ with the current global semantic features $f_{gv}^t$ and $f_{gl}^t$ respectively. This process yields more comprehensive visual and textual memory information, $H_v^{t-1} \in \mathbb{R}^{(L_m+1) \times D}$ and $H_l^{t-1} \in \mathbb{R}^{(L_m+1) \times D}$.

$$H_v^{t-1} = [^*M_v^{t-1}; f_{gv}^t] = [m_v^1; \ldots; m_v^{L_m}; f_{gv}^t], \tag{1}$$

$$H_l^{t-1} = [^*M_l^{t-1}; f_{gl}^t] = [m_l^1; \ldots; m_l^{L_m}; f_{gl}^t]. \tag{2}$$

where $^*$ denotes extracting all elements in the set, and [;] indicates concatenation along the first dimension. Following this, we introduce the memory query to associate long-term memory with the current information, enabling the generated short-term memory to consider both historical and current features simultaneously.

Considering the remarkable capability of the transformer network in feature interaction [45], we design the short-term memory generation (SMG) layer by modifying it. In each SMG layer, the memory query sequentially interacts with long-term memory information and corresponding modality features to generate short-term memory. Taking the visual branch as an example, the computational process of the visual query $q_v^0 \in \mathbb{R}^{1 \times D}$ is as follows:

$$q_{v'}^k = Norm(q_v^k + \Phi_{CA}(q_v^k, H_v^{t-1})), \tag{3}$$

$$q_{v''}^k = Norm(q_{v'}^k + \Phi_{CA}(q_{v'}^k, [f_z^t; f_x^t])), \tag{4}$$

$$q_v^{k+1} = Norm(q_{v''}^k + FFN(q_{v''}^k)). \tag{5}$$

Here, $\Phi_{CA}(\cdot, \cdot)$ denotes the cross-attention operation where the first element serves as $Q$ and the second element serves to obtain $K$ and $V$ [46]. $Norm$ represents the layer normalization operation and $FFN$ denotes the feed-forward network. $q_v^k$ denotes the query after being processed by the $k$-th layer. For brevity, we omit the positional encoding.

After passing through $L_v$ stacked SMG layers, we obtain the visual short-term memory $m_v^t = q_v^{L_v}$. Similarly, we utilize the textual query $q_l^0 \in \mathbb{R}^{1 \times D}$ to obtain the textual short-term memory $m_l^t = q_v^{L_l}$.

$$q_{l'}^k = Norm(q_l^k + \Phi_{CA}(q_l^k, H_l^{t-1})), \tag{6}$$

$$q_{l''}^k = Norm(q_{l'}^k + \Phi_{CA}(q_{l'}^k, [f_l^t; f_x^t])), \tag{7}$$

$$q_l^{k+1} = Norm(q_{l''}^k + FFN(q_{l''}^k)). \tag{8}$$

### 3.3.2 The Generation and Fusion of the Adaptive Prompts

Based on the acquired short-term memory (*i.e.*, $m^t = (m_v^t, m_l^t)$), on one hand, we feed it into MSM to update the long-term memory (see Sec. 3.4). On the other hand, we utilize it to modulate the long-term memory represented by the initial visual-textual prompts, enabling them to adapt to the

variations of the target. For the textual branch, we first utilize $m_l^t$ to modulate the initial textual feature $f_l^t$, then use the adjusted textual feature to guide the search feature $f_x^t$.

$$f_{al}^t = Trans_{Dec}(f_l^t, m_l^t), \tag{9}$$

$$f_{xl}^t = Trans_{Dec}(f_x^t, f_{al}^t). \tag{10}$$

Where $Trans_{Dec}$ denotes the vanilla transformer decoder layer [46] (see Sec. A.2 for detailed computation process). $f_{xl}^t$ represents the search features fused with the adaptive textual prompts $f_{al}^t$.

For the visual branch, we adopt a similar approach. As shown in Eq.11 and Eq.12, we ultimately obtain the search feature $f_{xlv}^t$ integrated with adaptive visual and textual prompts, simplified as $f_r^t$. We treat $f_r^t$ as the target-related features fed into the prediction head.

$$f_{av}^t = Trans_{Dec}(f_z^t, m_v^t), \tag{11}$$

$$f_r^t = f_{xvl}^t = Trans_{Dec}(f_x^t, f_{av}^t). \tag{12}$$

### 3.4 Memory Storage Module

In this section, we will introduce the construction of long-term memory $M^t$. Given a video sequence of length $T$, MIM generates a short-term memory $m^t$ at each time step. Typically, due to the computational burden of integrating all short-term memories, $L_m$ is much smaller than $T$. Therefore, the key to long-term memory storage is to design a short-term memory selection mechanism.

Prior trackers, *e.g.*, JointNLT and AQATrack, commonly use the *sliding window* approach to store temporal information, which can be considered a form of long-term memory. Specifically, when $t > L_m$, short-term memories are selected from the most recent $L_m$ frames (*i.e.*, from $t - L_m$ to $t-1$) for storage. While achieving some effectiveness, the redundancy of video frames leads to redundant memories being stored, and it can only consider a limited temporal range. The CLS theory refers to the process of integrating short-term memory into long-term memory as system consolidation [24]. Relevant studies [33, 23] indicate the long-term memory system primarily stores short-term memory conducive to generalization. We adhere to this principle to propose a novel long-term memory storage method called *section-top*. It first uniformly divides past time intervals into $L_m$ sections and then stores the most representative short-term memory within each section.

To establish a short-term memory selection criterion, we design a confidence prediction module to obtain the confidence score $p_c^t \in [0, 1]$ corresponding to the tracking result, which will be introduced in Sec. 3.5. Intuitively, a higher confidence score suggests that the current tracking result is more conducive for subsequent tracking and should thus be stored. The thorough storage process is provided in Sec. A.3. Compared to the *sliding window*, this method can consider a longer temporal range. Additionally, the stored memories are not adjacent to each other, which reduces redundancy.

### 3.5 Prediction Head and Loss

Based on the target-related search feature $f_r^t \in \mathbb{R}^{N_x \times D}$, the prediction head is used to predict the final bounding box $b^t$ and its corresponding confidence score $p_c^t$. We employ a classic CNN-based tracking head [41, 40]. First, $f_r^t$ is transformed into a 2D spatial feature map. Subsequently, after passing through the stacked Conv-BN-ReLU layers, we obtain a classification score map $P \in [0, 1]^{1 \times H_s \times W_s}$, the size of the bounding box $B \in [0, 1]^{2 \times H_s \times W_s}$, and the offset size $O \in [0, 1)^{2 \times H_s \times W_s}$. Based on these features, we predict the centroid position and scale of the target, yielding the predicted bounding box $b^t$. Additionally, we employ an additional CNN-based branch to predict $p_c^t$.

For the $b^t$, we employ the focal loss $L_{cls}$ [47], $L_1$ loss, and the generalized IoU loss $L_{iou}$ [48] for supervision, which are widely used in tracker design. Regarding the confidence score $p_c^t$, we first calculate the IoU value between the tracking result and the ground truth, and utilize the $L_2$ loss between IoU and $p_c^t$ for supervision. The overall loss function is formulated as follows:

$$L_{all} = \lambda_{L_{cls}} L_{cls} + \lambda_{iou} L_{iou} + \lambda_{L_1} L_1 + \lambda_{L_2} L_2, \tag{13}$$

where $\lambda_{L_{cls}} = 1$, $\lambda_{iou} = 2$, $\lambda_{L_1} = 5$ and $\lambda_{L_2} = 1$ are the regularization parameters.

Table 1: Comparison with state-of-the-arts on four popular benchmarks: MGIT [5], TNL2K [26], LaSOT [27] and LaSOT$_{ext}$ [28]. The best two results are highlighted in red and blue, respectively.

| Method | MGIT (Action) | | | TNL2K | | | LaSOT | | | LaSOT$_{ext}$ | | |
|---|---|---|---|---|---|---|---|---|---|---|---|---|
| | AUC | P$_{Norm}$ | P | AUC | P$_{Norm}$ | P | AUC | P$_{Norm}$ | P | AUC | P$_{Norm}$ | P |
| Wang [50] | - | - | - | - | - | - | 27.7 | - | 30.4 | - | - | - |
| Feng [51] | - | - | - | 25.0 | 34.0 | 27.0 | 50.0 | - | 56.0 | - | - | - |
| Feng [52] | - | - | - | 25.0 | 33.0 | 27.0 | 35.0 | - | 35.0 | - | - | - |
| GTI [29] | - | - | - | - | - | - | 47.8 | - | 47.6 | - | - | - |
| TNL2K-II [26] | - | - | - | 42.0 | 50.0 | 42.0 | 51.3 | - | 55.4 | - | - | - |
| SNLT [11] | 3.6 | 22.6 | 0.4 | - | - | - | 54.0 | 63.6 | 57.4 | - | - | - |
| Li [12] | - | - | - | 44.0 | 52.0 | 45.0 | 53.0 | 56.0 | - | - | - | - |
| VLTTT [13] | 46.8 | 60.2 | 31.8 | 54.7 | 71.8 | 55.3 | 67.3 | 80.2 | 71.5 | 48.4 | 59.9 | 54.3 |
| TransVLT [53] | - | - | - | 56.0 | 61.7 | - | 66.4 | - | 70.8 | - | - | - |
| JointNLT [14] | 61.0 | 78.6 | 44.5 | 56.9 | 73.6 | 58.1 | 60.4 | 69.4 | 63.6 | - | - | - |
| TransNLT [15] | - | - | - | 57.0 | 75.0 | 57.0 | 60.0 | - | 63.0 | - | - | |
| DecoupleTNL [32] | - | - | - | 56.7 | - | 56.0 | 71.2 | - | 75.3 | - | - | - |
| MMTrack [16] | - | - | - | 58.6 | 75.2 | 59.4 | 70.0 | 82.3 | 75.7 | 49.4 | 59.9 | 55.3 |
| QueryNLT [54] | - | - | - | 56.9 | 73.6 | 58.1 | 59.9 | 69.6 | 63.5 | - | - | - |
| **Ours** | 69.4 | 81.3 | 63.7 | 63.3 | 80.9 | 67.4 | 72.9 | 85.7 | 80.5 | 52.1 | 63.3 | 59.8 |

## 4 Experiments

### 4.1 Implementation Details

We use RoBERTa-Base [44] as our text encoder and HiViT-Base [42, 39, 40] as our vision encoder, with the token dimension $D$ set to 512. The sizes of template patches and search images are $192 \times 192$ and $384 \times 384$, respectively. For the acquisition of short-term memory, both the visual and textual branches consist of three SMG layers. For the generation and fusion of adaptive prompts, all $Trans_{Dec}$ operations are conducted using a single transformer decoder layer. Additionally, in the memory storage module, the default length for long-term memory is set to eight.

We use the training splits of LaSOT [27], TNL2K [26], RefCOCOg [49], and OTB99-Lang [1] to train our model. Each training sample consists of a text description, along with one template patch and eight search frames from the same video sequence. Utilizing the text description and template patch as prompts, we iteratively train the model by selecting one search image at a time. We employ the AdamW to optimize the network parameters and conduct a total of 200 training epochs. 20,000 image pairs are randomly sampled in each epoch. The model is trained on a server with four A5000 GPUs and tested on an RTX-3090 GPU. The tracking speed is about 32 FPS.

### 4.2 Comparison with State-of-the-art

We evaluate MemVLT on four benchmarks, including MGIT [5], TNL2K [26], LaSOT [27] and LaSOT$_{ext}$ [28]. MemVLT is compared with existing state-of-the-art (SOTA) VLTs, which share the similar task setting and training datasets configuration to ensure a fair comparison.

**MGIT.** MGIT is a novel large-scale benchmark [2, 5] specifically tailored for the VLT task. Each sequence contains complex spatio-temporal causal relationships and is annotated with language descriptions at three levels of granularity: action, activity, and story [5]. As shown in Tab. 1, MemVLT demonstrates superior performance compared to other VLTs at the action granularity. Particularly, MemVLT excels over the SOTA tracker JointNLT [14], surpassing it by 8.4%, 2.7%, and 19.2% in area under the curve (AUC), normalized precision (P$_{Norm}$), and precision score (P), respectively. Although JointNLT is equipped with a temporal module, this temporal information does not interact with the initial prompts. In other words, the initial static prompts still dominate the tracking process. These results highlight that the utilization of adaptive prompts plays a crucial role in complex scenarios. Additionally, our model also achieves optimal performance under the activity and story text settings (see Tab. A1).

Table 3: Ablation study on important model components. The best results are shown in red.

| # | $Vision_{adap}$ | $Text_{adap}$ | AUC | $P_{Norm}$ | P |
|---|---|---|---|---|---|
| 1 | | | 59.0 | 77.6 | 62.4 |
| 2 | ✓ | | 62.3 | 79.5 | 66.2 |
| 3 | | ✓ | 62.4 | 79.9 | 66.5 |
| 4 | ✓ | ✓ | 63.3 | 80.9 | 67.4 |

Table 4: Comparison of different long-term memory storage methods.

| Method | AUC | $P_{Norm}$ | P |
|---|---|---|---|
| *sliding window* | 62.3 | 79.8 | 66.6 |
| *top-L* | 62.5 | 80.0 | 66.8 |
| *section-L* | 62.8 | 80.7 | 66.9 |
| *section-top* | 63.3 | 80.9 | 67.4 |

**TNL2K.** TNL2K is designed for the VLT task, and the introduction of attributes such as "adversarial samples", "modality switch" and "cross-camera" significantly adds to the challenges[26]. In Tab. 1, our proposed framework demonstrates superior performance compared to existing VLTs. When compared with SOTA tracker MMTrack [16], which doesn't utilize the temporal information, our approach achieves gains of 4.7%, 5.7%, and 8.0% in AUC, $P_{Norm}$, and P, respectively. These results underscore the effectiveness of leveraging memory information for providing adaptive prompts.

**LaSOT and LaSOT$_{ext}$.** LaSOT and LaSOT$_{ext}$ are extensions of traditional long-term visual tracking benchmarks [27, 55] by adding language annotations. In addition to the challenges in long-term tracking, sequences in LaSOT$_{ext}$ also include many similar distractors, further complicating the tracking task. Nevertheless, our method still achieves outstanding performance, as demonstrated in Tab. 1. For instance, our model outperforms the SOTA algorithms in LaSOT and LaSOT$_{ext}$ by 1.7% and 2.7% in terms of AUC, respectively. The strong performance across multiple benchmarks also reflects the generalization capability of MemVLT to diverse video environments and corresponding linguistic annotation styles.

Table 2: Results of efficiency analysis.

| Model | Params | Speed | AUC | P |
|---|---|---|---|---|
| JointNLT | 153M | 31FPS | 56.9 | 58.1 |
| MMTrack | 177M | 37FPS | 58.6 | 58.1 |
| MemVLT | 175M | 32FPS | 63.3 | 67.4 |

**Efficiency analysis.** In Tab. 2, we compare MemVLT with the two latest VLTs in terms of efficiency (Params and Speed) and performance (AUC and P on TNL2K). Compared to JointNLT [14], which only introduces temporal features in the decoder phase, MemVLT fully leverages memory information during the feature fusion stage. Despite the increased number of parameters, the model performance is significantly improved without compromising tracking speed. As for MMTrack [16], the absence of temporal modeling accelerates its forward process but limits tracking performance. These results underscore the trade-off between efficiency and performance achieved by MemVLT.

### 4.3 Ablation Study

In this section, we conduct comprehensive ablation studies on the TNL2K [26] benchmark.

**Study on important model components.** The core insight of our proposed model lies in the modeling and utilization of memory information, which adjusts initial static visual-textual prompts to adapt to the dynamic changes of the target. To investigate the effect of different model components, we conduct ablation analyses based on whether the incorporation of memory information is utilized to generate the adaptive visual or textual prompts, corresponding to the $Vision_{adap}$ and $Text_{adap}$ entries in Tab. 3. Tab. 3 (#1) presents the performance without utilizing the memory information. Comparing it with the results in Tab. 3 (#2, #3), we observe that utilizing memory information in any single modality can enhance performance. For example, adaptive visual or textual prompts lead to a 3.3% and 3.4% increase in AUC, respectively. Additionally, Tab. 3 (#4) further demonstrates that, under the joint influence of adaptive visual-textual prompts, the model achieves optimal performance.

**Study on memory interaction.** The MIM plays a crucial role in the forward process, and we conduct ablation analysis on specific model designs. Tab. 5 (#2) indicates that global visual and textual tokens (*i.e.*, $f_{gv}^t$ and $f_{gl}^t$) are excluded from the construction of memory information. Comparing with the baseline (Tab. 5 (#1)), it is evident that the inclusion of global tokens enhances tracking performance. Tab. 5 (#3) shows that directly integrating search features with short-term memories significantly

Table 5: Ablation study on memory interaction.

| # | Setting | AUC | $P_{Norm}$ | P |
|---|---------|-----|------------|---|
| 1 | *baseline* | 63.3 | 80.9 | 67.4 |
| 2 | *remove global tokens* | 62.1 | 79.2 | 65.9 |
| 3 | *remove adaptive prompts* | 60.9 | 79.0 | 65.2 |

Table 6: Generalizability analysis on SOT.

| # | Setting | AUC | $P_{Norm}$ | P |
|---|---------|-----|------------|---|
| 1 | Naive | 57.4 | 74.9 | 60.6 |
| 2 | + MIM | 58.3 | 76.1 | 61.0 |
| 2 | + MSM | 59.6 | 76.8 | 62.3 |

reduces the model's performance. This indicates the importance of the adaptive prompts generation process.

**Study on long-term memory lengths.** we employ a buffer $M^t$ with a length of $L_m$ to represent long-term memory. Here, we analyze the model's performance under different buffer lengths. As shown in Tab. 7, we observe that as the buffer length increases, the model's performance initially rises and then stabilizes. Notably, the model achieves optimal performance when the buffer length is set to 8.

**Study on long-term memory storage method.** For our proposed *section-top* method, we compare it with other storage methods. As shown in Tab. 4, *sliding window* is commonly adapted by recent trackers involving temporal saving [14, 30, 31, 32]; *top-L* prioritizes storing the top $L$ short-term memories with the highest confidence; *section-L* divides historical memory into $L$ sections, with each section storing the latest short-term memory. It can be observed that the performance of *top-L* is superior to *sliding window*, highlighting the significance of storing important short-term memories. Additionally, *section-L* outperforms *top-L*, indicating the benefits of considering a longer temporal range. By combining the advantages of *section-L* and *top-L*, *section-top* achieves optimal performance.

Table 7: Ablation study on long-term memory lengths.

| Length | AUC | $P_{Norm}$ | P |
|--------|-----|------------|---|
| 1 | 62.1 | 79.2 | 65.9 |
| 2 | 62.5 | 79.9 | 66.5 |
| 4 | 62.8 | 80.2 | 66.9 |
| 6 | 63.1 | 80.7 | 67.5 |
| 8 | 63.3 | 80.9 | 67.4 |
| 10 | 63.1 | 80.8 | 67.5 |

**Generalizability of our memory mechanism.** Given that our proposed memory mechanism modeling significantly enhances the performance of vision-language trackers, a natural question arises: can this memory mechanism generalize to vision-only taskers? To explore this, it is necessary to conduct relevant experimental analysis. Specifically, we remove the text-related components from MemVLT, seamlessly converting it into a vision-only tracker. As shown in Tab. 6, our proposed Memory Interaction Module (MIM) and Memory Storage Module (MSM) progressively improve the performance of the vision-only tracker, demonstrating the strong generalizability of our method.

For more detailed implementation specifics and further comparative analysis, please refer to Sec. B and Sec. C.

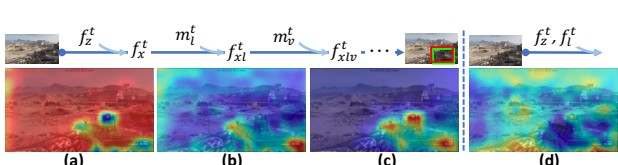
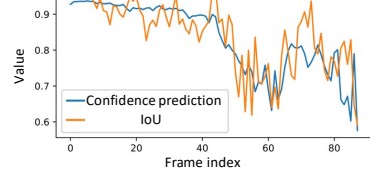

Figure 3: (a)-(c): Heatmaps obtained during the forward process of MemVLT, integrating various adaptive prompts. (d): Heatmap guided solely by the initial fixed prompts. The above process diagrams depict the types and sequence of feature integration. We also illustrate the tracked result bbox and groundtruth bbox. Better viewed with zoom-in.

Figure 4: Comparison between confidence prediction and IoU values (taking the "advSamp_INF_bus6" sequence as an Example).

### 4.4 Qualitative Analysis.

Fig. 3 (a)-(c) depict the variations of heatmaps over the search region during the forward process. Through the integration of adaptive prompts, the model ultimately directs its focus towards the tracking target. Compared to merely fusing initial fixed multimodal prompts (Fig. 3 (d)), MemVLT demonstrates improved capability in scenarios where the target state deviates from the given static prompts.

Taking the "advSamp_INF_bus6" sequence from TNL2K as an example, we showcase the model's confidence prediction scores against the actual IoU values in Fig. 4. It can be observed that their variations align closely, indicating the effectiveness of our confidence prediction module.

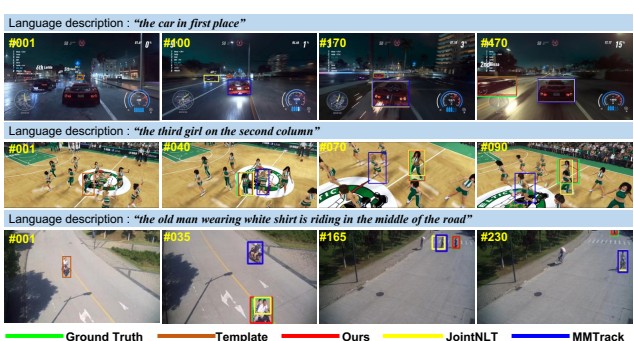

Figure 5: Qualitative comparison results of our tracker with other two latest VLTs (*i.e.*, JointNLT [14] and MMTrack [16]) on three challenging sequences from TNL2K [26] benchmark. The first column indicates the provided initial template information. Better viewed in color with zoom-in.

This facilitates the utilization of the confidence prediction score as criteria for selecting short-term memory.

As shown in Fig. 5, we visualize the tracking results of our model and the previous two SOTA models on three challenging sequences from TNL2K [26]. In these sequences, the scenes contain distractors, and the state of the target undergoes significant changes. It is evident that our model exhibits greater robustness [56] compared to others. This validates that our adaptive prompts contribute to addressing these challenges, further demonstrating the efficacy of our proposed model.

## 5    Conclusion

We propose a novel vision-language tracker, MemVLT, which models memory mechanisms to provide adaptive multimodal prompts for tracking guidance. Drawing from the Complementary Learning Systems theory, we emphasize the importance of storing and interacting between short-term and long-term memories for generalized adaptation. Therefore, we incorporate memory storage and memory interaction modules. By comprehensively leveraging memory information to generate adaptive prompts, MemVLT provides consistent references for dynamically changing targets, thus achieving effective tracking performance. Extensive experiments demonstrate that our method achieves new state-of-the-art performance on four widely used benchmarks, showcasing its generalization ability across various video environments and linguistic annotation styles.

**Limitations.**    Our proposed method leverages learnable queries to implicitly construct the relationship between historical target information and the current situation, resulting in short-term memory representations of the target. The notable results obtained demonstrate the effectiveness of this memory modeling mechanism. However, these memory representations lack explicit supervision, which diminishes their interpretability. To address this limitation, we believe that incorporating tracking result data (such as target and background information at various time steps) into the memory representation could be beneficial. This integration could enhance the comprehensiveness of the memory representations and facilitate the design of diverse loss functions for explicit supervision. We consider this approach a promising direction for our future research.

## Acknowledgments and Disclosure of Funding

This work is jointly supported by the National Science and Technology Major Project (No.2022ZD0116403), the National Natural Science Foundation of China (No.62176255), and the Strategic Priority Research Program of Chinese Academy of Sciences (No.XDA27010201).

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

# Appendices

## A  More Details on the MemVLT Model

In the following sections, we will provide comprehensive details on the implementation of MemVLT.

### A.1  Input Encoder

**Vision Encoder.** As discussed in Sec. 3.2, our vision encoder is based on HiVit [42], which is constructed by stacking a series of transformer network layers. To obtain global semantic features, we introduce the [CLS] token [43] in the last two layers. Specifically, the initial template and search image tokens are concatenated and jointly modeled, aligning with the existing one-stream network architecture [41]. After obtaining the initial encoded template-search features, we introduce the [CLS] token. The [CLS] token is then concatenated with the obtained template-search features. This concatenated feature is fed into the last two layers for further feature extraction and relationship modeling. Finally, we obtain the search feature $\check{f}_x^t \in \mathbb{R}^{N_x \times D}$, template feature $f_z^t \in \mathbb{R}^{N_z \times D}$, and the global visual feature $f_{gv}^t \in \mathbb{R}^{1 \times D}$.

**Text Encoder.** We employ the RoBERTa [44] model as our text encoder. Existing VLTs typically set a maximum text truncation length, *e.g.*, JointNLT supports a maximum text length of 40. However, for datasets providing detailed descriptions of target states, such as the MGIT dataset[5], the text length often exceeds this threshold. Simply truncating the text would diminish its utility for the tracker. Therefore, we do not set a maximum text truncation length and use sinusoidal position encoding to accommodate variable text lengths.

### A.2  Memory Interaction Module

As shown in Fig. 2, the Memory Interaction module mainly consists of two types of networks: the short-term memory generation (SMG) layer and the transformer decoder layer. To facilitate a better understanding of their specific computational processes, we illustrate their diagram in Fig. A1.

For the transformer decoder layer [46], we illustrate the detailed computation process corresponding to $Trans_{dec}$ (as shown in Eq. 9, Eq. 10, Eq. 11, Eq. 12) as follows:

$$x' = Norm(x + \Phi_{CA}(x, y)), \tag{A1}$$

$$z = Norm(x' + FFN(x')). \tag{A2}$$

Here, $\Phi_{CA}(\cdot, \cdot)$ denotes the cross-attention operation where the first element serves as $Q$ and the second element serves to obtain $K$ and $V$ [46]. $Norm$ represents the layer normalization operation and $FFN$ denotes the feed-forward network. For brevity, we omit the positional encoding.

Compared to the standard transformer decoder layer [46], we omit the initial self-attention operation. This helps reduce computational overhead.

### A.3  Memory Storage Module

For the *section-top* long-term memory storage method proposed in this work, the basic approach is to first uniformly divide past time intervals into $L_m$ sections and then select the most representative short-term memory within each section to save.

To achieve this goal, we introduce two buffers, $\mathcal{B}^t$ and $\mathcal{B}_c^t$, which are used to store all memory units up to time $t$ and their corresponding confidence scores, respectively. We initialize $\mathcal{B}^t$ and $\mathcal{B}_c^t$ as empty sets. For the long-term memory information $M^t$ that we aim to obtain, we initialize each memory unit with $\mathbf{0} \in \mathbb{R}^{1 \times D}$. At time $t$ ($t \geq 1$), the detailed computation process is illustrated in Algorithm 1.

### A.4  Prediction Head

The prediction head is used to predict the final bounding box $b^t$ and its corresponding confidence score $p_c^t$. We employ a CNN-based tracking head [41, 40], which is widely adopted in tracker design. Firstly, the target-related search feature $f_r^t \in \mathbb{R}^{N_x \times D}$ is transformed into a 2D spatial feature map. Subsequently, after passing through $L_h$ stacked Conv-BN-ReLU layers, we obtain a classification

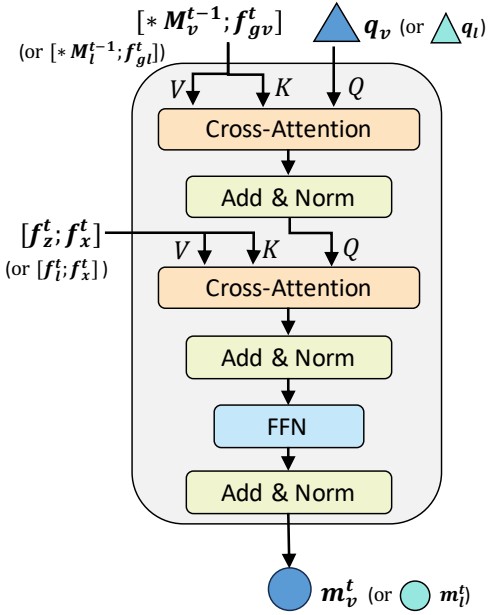
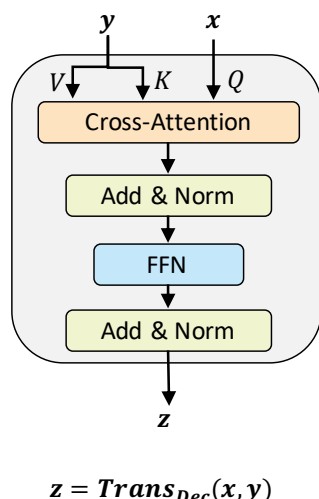

$$z = Trans_{Dec}(x, y)$$

(a) Short-term Memory Generation Layer  (b) Transformer Decoder Layer

Figure A1: (a) The diagram of the short-term memory generation (SMG) Layer. Depending on the input modality data, this module is capable of generating short-term memories for either visual or textual inputs. (b) The diagram of the transformer decoder layer. It primarily consists of a cross-attention operation and a fully connected operation.

score map $P \in [0, 1]^{1 \times H_s \times W_s}$, the size of the bounding box $B \in [0, 1]^{2 \times H_s \times W_s}$, and the offset size $O \in [0, 1)^{2 \times H_s \times W_s}$.

Based on these features, we calculate the tracking bounding box $b^t$ and the confidence score $p_c^t$ through two branches. On one hand, the position with the highest classification score is considered to be the target position, *i.e.*, $(x_d, y_d) = \arg \max_{(x,y)} P_{xy}$. The final target bounding box is obtained as:

$$(x, y, w, h) = (x_d + O(0, x_d, y_d), y_d + O(1, x_d, y_d), S(0, x_d, y_d), S(1, x_d, y_d)). \quad (A3)$$

On the other hand, we employ an additional branch to predict $p_c^t$. We first concatenate $P$, $B$, and $O$ along the first dimension. Then, this concatenated feature is processed through stacked Conv-BN-ReLU layers. Finally, the processed features are flattened and passed through a softmax function to produce a regression prediction value within the range [0,1].

## B   More Implementation Details on Experimental Analysis

In the following sections, we will present the implementation details of our experimental analysis, including the consistency analysis between search images and prompts, as well as the implementation methods for various model variants in the ablation study.

### B.1   Consistency Analysis

As shown in Fig. 1 (a), to quantitatively analyze the consistency between multimodal prompts and the search target, we calculate and plot the consistency curves. For determining the consistency between the search target and the language description, we manually evaluate whether the target state matches the language description, assigning a consistency value of 1 or 0 accordingly. For computing the consistency between the search target and the template patch, we first crop the search target from the search image based on the groundtruth bounding box. Then, we input both the cropped target

**Algorithm 1** *Section-Top* Long-term Memory Storage Algorithm

---

**Input**: $\mathcal{B}^{t-1}$, $\mathcal{B}_c^{t-1}$, $M^{t-1}$, $m^t$, $p_c^t$ at time $t$
**Output**: $M^t$

1: % save the latest memory unit and its corresponding confidence score.
2: $\mathcal{B}^t = \mathcal{B}^{t-1}.append(m^t)$, $\mathcal{B}_c^t = \mathcal{B}_c^{t-1}.append(p_c^t)$
3: $M^t = M^{t-1}$
4: **if** $t \leq L_m$ **then**
5:    % select the latest memory to save.
6:    $M^t[t] = m^t$
7: **else**
8:    $t_m = t \bmod L_m$, $t_d = \lfloor \frac{t}{L_m} \rfloor$
9:    $I_{\text{start}} = (t_d + 1) \times (t_m - 1) + 1$, $I_{\text{end}} = I_{\text{start}} + t_d$
10:    % select most representative memory unit in each section
11:    **for** $i$ **in** $t_m$ **to** $L_m$ **do**
12:       $Sec_m = \mathcal{B}^t[I_{\text{start}} : I_{\text{end}}]$, $Sec_c = \mathcal{B}_c^t[I_{\text{start}} : I_{\text{end}}]$
13:       $I_{\max} = \text{MaxIndex}(Sec_c)$
14:       $M^t[i] = \mathcal{B}^t[I_{\max}]$
15:       $I_{\text{start}} = I_{\text{end}} + 1$, $I_{\text{end}} = I_{\text{start}} + t_d - 1$
16:    **end for**
17: **end if**
18: **return** $M^t$

---

and the template patch into a pretrained ResNet backbone network for feature encoding. Finally, the consistency value is obtained by calculating the cosine similarity between the encoded features.

## B.2 Effectiveness Analysis of Different Model Components

Corresponding to Tab. 3, we conduct ablation analyses to investigate the effect of different model components, based on whether the incorporation of memory information is utilized to generate adaptive visual or textual prompts. When adaptive visual prompts are not used (i.e., *Vision*$_{adap}$ is not selected), the MIM module neither generates short-term visual memory nor subsequent adaptive visual prompts. Instead, the search feature interacts only with the initial template feature, replacing Eq. 3, Eq. 4, Eq. 5, Eq. 9 and Eq. 10 with Eq. A4. Additionally, to ensure a fair comparison, we set the number of network layers used in Eq. A4 to be the sum of the network layers in Eq. 3, Eq. 4, Eq. 5, Eq. 9 and Eq. 10.

$$f_{xl}^t = Trans_{Dec}(f_x^t, f_l^t). \tag{A4}$$

Additionally, not selecting *Vision*$_{adap}$ means that no short-term visual memory is stored in the MSM module. Similarly, by not selecting *Text*$_{adap}$, we can achieve this by replacing Eq. 6, Eq. 7, Eq. 8, Eq. 11 and Eq. 11 with the following equation:

$$f_r^t = Trans_{Dec}(f_{xl}^t, f_z^t). \tag{A5}$$

## B.3 Variants of Memory Interaction Module

Corresponding to Tab. 5, we conduct ablation analyses on the relevant model designs in MIM. The "*remove global tokens*" setting means that the global visual and textual tokens (i.e., $f_{gv}^t$ and $f_{gl}^t$) are excluded from the construction of memory information. Specifically, the modified expressions for $H_v^{t-1}$ and $H_l^{t-1}$ are as follows:

$$H_v^{t-1} = [^*M_v^{t-1}] = [m_v^1; \ldots; m_v^{L_m}]. \tag{A6}$$

$$H_l^{t-1} = [^*M_l^{t-1}] = [m_l^1; \ldots; m_l^{L_m}]. \tag{A7}$$

The "*remove adaptive prompts*" setting means directly integrating search features with short-term memories, which removes the generation of adaptive prompts and their subsequent fusion with the

---

**Algorithm 2** *Sliding Window* Long-term Memory Storage Algorithm

---

**Input**: $M^{t-1}, m^t, p_c^t$ at time $t$
**Output**: $M^t$

1:  $M^t = M^{t-1}$
2:  **if** $t \leq L_m$ **then**
3:     % select the latest memory to save.
4:     $M^t[t] = m^t$
5:  **else**
6:     % sliding window process.
7:     $M^t[1 : (L_m - 1)] = M^t[2 : L_m]$
8:     $M^t[L_m] = m^t$
9:  **end if**
10: **return** $M^t$

---

---

**Algorithm 3** *Section-L* Long-term Memory Storage Algorithm

---

**Input**: $\mathcal{B}^{t-1}, M^{t-1}, m^t, p_c^t$ at time $t$
**Output**: $M^t$

1:  % save the latest memory unit.
2:  $\mathcal{B}^t = \mathcal{B}^{t-1}.append(m^t)$
3:  $M^t = M^{t-1}$
4:  **if** $t \leq L_m$ **then**
5:     % select the latest memory to save.
6:     $M^t[t] = m^t$
7:  **else**
8:     $t_m = t \bmod L_m, t_d = \lfloor \frac{t}{L_m} \rfloor$
9:     $I_{start} = (t_d + 1) \times (t_m - 1) + 1, I_{end} = I_{start} + t_d$
10:    % select the latest memory unit in each section
11:    **for** $i$ **in** $t_m$ **to** $L_m$ **do**
12:       $M^t[i] = \mathcal{B}^t[I_{end}]$
13:       $I_{start} = I_{end} + 1, I_{end} = I_{start} + t_d - 1$
14:    **end for**
15: **end if**
16: **return** $M^t$

---

search features. For the visual branch, we replace Eq. 9 and Eq. 10 with Eq. A8. Similarly, for the textual branch, we replace Eq. 11 and Eq. 12 with Eq. A9

$$f_{xl}^t = Trans_{Dec}(f_x^t, m_l^t), \tag{A8}$$

$$f_r^t = Trans_{Dec}(f_{xl}^t, m_v^t). \tag{A9}$$

Additionally, to ensure a fair comparison, we set the number of network layers used in Eq. A8 to be the sum of the network layers in Eq. 9 and Eq. 10, and set the number of network layers used in Eq. A9 to be the sum of the network layers in Eq. 11 and Eq. 12.

### B.4   Variants of Memory Storage Module

Corresponding to Tab. 4, we conduct ablation analyses on different long-term memory storage methods in MSM. In addition to the *section-top* long-term memory storage method shown in Algorithm 1, we also present the implementation processes of three other storage methods in Algorithm 2, Algorithm 3, and Algorithm 4. For *top-L* method, we introduce a new variable $M_c^t$, which stores the confidence store corresponding to each memory unit in $M^t$.

**Algorithm 4** *Top-L* Long-term Memory Storage Algorithm
___
**Input**: $\mathcal{M}_c^{t-1}$, $M^{t-1}$, $m^t$, $p_c^t$ at time $t$
**Output**: $\tilde{M}^t$
  1: $M^t = M^{t-1}$, $M_c^t = M_c^{t-1}$
  2: **if** $t \le L_m$ **then**
  3:    % select the latest memory to save.
  4:      $M^t[t] = m^t$
  5:      $M_c^t[t] = p_c^t$
  6: **else**
  7:      $I_{min} = MinIndex(M_c^t)$, $p_{min} = Min(M_c^t)$
  8:      **if** $t \le L_m$ **then**
  9:        $I_{max} = MaxIndex(Sec_c)$
 10:        $M^t[I_{min}] = m^t$
 11:        $M_c^t[I_{min}] = p_c^t$
 12:      **end for**
 13: **end if**
 14: **return** $M^t$

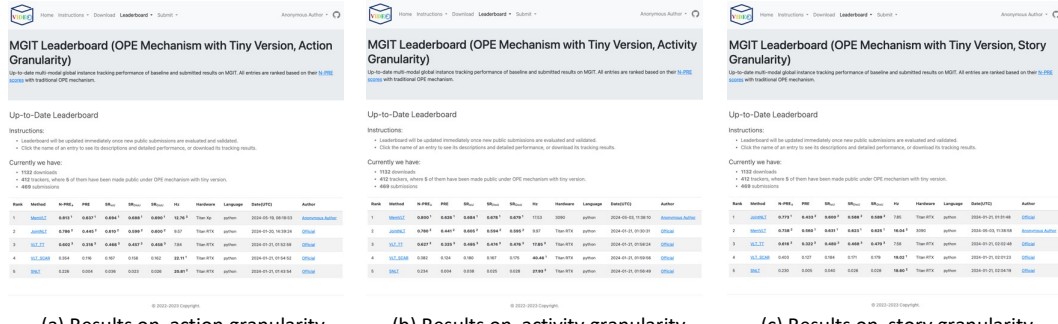

(a) Results on action granularity  (b) Results on activity granularity  (c) Results on story granularity

Figure A2: Screenshot of the Latest MGIT Leaderboard for action granularity, activity granularity, and story granularity. Better viewed with zoom-in.

## C  Additional Experimental Analysis

### C.1  Experimental Results of All Granularities on MGIT

MGIT is the latest large-scale benchmark specifically tailored for the VLT task, focusing on long-term challenges with complex spatio-temporal causal relationships. Each sequence is annotated with language descriptions at three levels of granularity: action, activity, and story [5]. Due to space constraints, we only present the experimental results for the action granularity in Tab. 1. We have anonymously submitted results for all three granularities on the official testing platform. As shown in Fig. A2, our model achieves state-of-the-art performance across nearly all metrics for all three granularities. Specifically, MemVLT surpasses the existing best results by 9%, 8%, and 3% in area under the curve for the three granularities, respectively. We expect that our proposed MemVLT can serve as a solid baseline.

### C.2  Comparison with Visual-only Trackers

In line with the prevailing paradigm of vision-language tracking models [13, 14, 16], we provide additional comparisons with visual-only trackers to comprehensively showcase the performance of our model. As shown in Tab. A1, we organize and present the results of visual-only tracking models to supplement Sec. 4.2.

**TNL2K.**  Despite being proposed for the VLT task, TNL2K has become a commonly used benchmark for the visual tracking task (as shown in Tab. A1). The primary intention of TNL2K is to exploit the complementarity of visual and textual modalities to achieve robust tracking performance,

Table A1: Comparison with state-of-the-arts on three popular benchmarks: TNL2K [26], LaSOT [27] and LaSOT$_{ext}$ [28]. The vision-only type of methods are evaluated by bounding box initialization, while the vision-language (VL) type of methods are evaluated by joint bounding box and natural language initialization. The best two results are highlighted in red and blue, respectively.

| Type | Method | TNL2K | | | LaSOT | | | LaSOT$_{ext}$ | | |
|---|---|---|---|---|---|---|---|---|---|---|
| | | AUC | P$_{Norm}$ | P | AUC | P$_{Norm}$ | P | AUC | P$_{Norm}$ | P |
| Vision-related | Mixformer [57] | - | - | - | 69.2 | 78.7 | 74.7 | - | - | - |
| | TransInMo [58] | 52.0 | 58.5 | 52.7 | 65.7 | 76.0 | 70.7 | - | - | - |
| | OSTrack-256 [41] | 54.3 | - | - | 69.1 | 78.7 | 75.2 | 47.4 | 57.3 | 53.3 |
| | OSTrack-384 [41] | 55.9 | - | - | 71.1 | 81.1 | 77.6 | 50.5 | 61.3 | 57.6 |
| | AiATrack [59] | - | - | - | 69.0 | 79.4 | 73.8 | 47.7 | 55.6 | 55.4 |
| | SimTrack [60] | - | - | - | 69.3 | 78.5 | - | - | - | - |
| | GRM [61] | - | - | - | 69.9 | 79.3 | 75.8 | - | - | - |
| | SeqTrack-B256 [62] | 54.9 | - | - | 69.9 | 79.7 | 76.3 | 49.5 | 60.8 | 56.3 |
| | SeqTrack-B384 [62] | 56.4 | - | - | 71.5 | 81.1 | 77.8 | 50.5 | 61.6 | 57.5 |
| | ARTrack-256 [63] | 57.5 | - | - | 70.4 | 79.5 | 76.6 | 46.4 | 56.5 | 52.3 |
| | ARTrack-384 [63] | 59.8 | - | - | 72.6 | 81.7 | 79.1 | 51.9 | 62.0 | 58.5 |
| | CiteTracker [64] | 57.7 | - | 59.6 | 69.7 | 78.6 | 75.7 | - | - | - |
| | DropTrack [65] | 56.9 | - | 57.9 | 71.8 | 81.8 | 78.1 | 52.7 | 63.9 | 60.2 |
| | ROMTrack-256 [66] | - | - | - | 69.3 | 78.8 | 75.6 | 48.9 | 59.3 | 55.0 |
| | ROMTrack-384 [66] | - | - | - | 71.4 | 81.4 | 78.2 | 51.3 | 62.4 | 58.6 |
| | F-BDMTrack-256 [67] | 56.4 | - | 56.5 | 69.9 | 79.4 | 75.8 | 47.9 | 57.9 | 54.0 |
| | F-BDMTrack-384 [67] | 57.8 | - | 59.4 | 72.0 | 81.5 | 77.7 | 50.8 | 61.3 | 57.8 |
| | EVPTrack-224 [39] | 57.5 | - | 58.8 | 70.4 | 80.9 | 77.2 | 48.7 | 59.5 | 55.1 |
| | EVPTrack-384 [39] | 59.1 | - | 62.0 | 72.7 | 82.9 | 80.3 | 53.7 | 65.5 | 61.9 |
| | ODTrack-B [38] | 60.9 | - | - | 73.2 | 83.2 | 80.6 | 52.4 | 63.9 | 60.1 |
| | ODTrack-L [38] | 61.7 | - | - | 74.0 | 84.2 | 82.3 | 53.9 | 65.4 | 61.7 |
| | AQATrack-256 [40] | 57.8 | - | 59.4 | 71.4 | 81.9 | 78.6 | 51.2 | 62.2 | 58.9 |
| | AQATrack-384 [40] | 59.3 | - | 62.3 | 72.7 | 82.9 | 80.2 | 52.7 | 64.2 | 60.8 |
| | ARTrackV2-256 [68] | - | - | - | 71.6 | 80.2 | 77.2 | 50.8 | 61.9 | 57.7 |
| | ARTrackV2-384 [68] | - | - | - | 73.0 | 82.0 | 79.6 | 52.9 | 63.4 | 59.1 |
| | HIPTrack [69] | - | - | - | 72.7 | 82.9 | 79.5 | 53.0 | 64.3 | 60.6 |
| | RTracker-L [70] | 60.6 | - | 63.7 | 74.7 | 84.5 | - | 54.9 | 65.5 | 62.7 |
| | OneTracker [71] | - | - | - | 70.5 | 79.9 | 76.5 | - | - | - |
| Vision-Language | Wang [50] | - | - | - | 27.7 | - | 30.4 | - | - | - |
| | Feng [51] | 25.0 | 34.0 | 27.0 | 50.0 | - | 56.0 | - | - | - |
| | Feng [52] | 25.0 | 33.0 | 27.0 | 35.0 | - | 35.0 | - | - | - |
| | GTI [29] | - | - | - | 47.8 | - | 47.6 | - | - | - |
| | TNL2K-II [26] | 42.0 | 50.0 | 42.0 | 51.3 | - | 55.4 | - | - | - |
| | SNLT [11] | - | - | - | 54.0 | 63.6 | 57.4 | - | - | - |
| | Li [12] | 44.0 | 52.0 | 45.0 | 53.0 | 56.0 | - | - | - | - |
| | VLT$_{TT}$ [13] | 54.7 | 71.8 | 55.3 | 67.3 | 80.2 | 71.5 | 48.4 | 59.9 | 54.3 |
| | TransVLT [53] | 56.0 | 61.7 | - | 66.4 | - | 70.8 | - | - | - |
| | JointNLT [14] | 56.9 | 73.6 | 58.1 | 60.4 | 69.4 | 63.6 | - | - | - |
| | TransNLT [15] | 57.0 | 75.0 | 57.0 | 60.0 | - | 63.0 | - | - | - |
| | DecoupleTNL [32] | 56.7 | - | 56.0 | 71.2 | - | 75.3 | - | - | - |
| | MMTrack [16] | 58.6 | 75.2 | 59.4 | 70.0 | 82.3 | 75.7 | 49.4 | 59.9 | 55.3 |
| | QueryNLT [54] | 56.9 | 73.6 | 58.1 | 59.9 | 69.6 | 63.5 | - | - | - |
| | **Ours** | 63.3 | 80.9 | 67.4 | 72.9 | 85.7 | 80.5 | 52.1 | 63.3 | 59.8 |

especially in challenging video sequences where the target state changes drastically [26]. However, recent vision-language trackers [14, 16] have shown inferior performance compared to vision-only trackers [38, 70]. In contrast, our model outperforms SOTA vision-language trackers and visual-only trackers. Specifically, we achieve improvements of 4.7 % in AUC over MMTrack [16] and 1.6 % in AUC over RTracker [38]. These results highlight the importance of the various adaptive prompts (visual and textual) we incorporated in addressing complex scenarios.

**LaSOT and LaSOT$_{ext}$.** As the mainstream benchmarks for visual tracking tasks, LaSOT [27] and LaSOT$_{ext}$ [28] are extended to the VLT task by incorporating language annotations. As shown

in Tab. A1, our proposed MemVLT outperforms existing vision-language trackers on these two benchmarks. Compared to SOTA tracker MMTrack [16], our model surpasses it by 2.9 % and 2.7 % in area under the curve , respectively. However, we also notice that the performance of MemVLT does not surpass existing SOTA visual-only trackers [38, 70]. We speculate that this may be related to the quality of some language descriptions on these two benchmarks. For instance, some descriptions may be too vague to effectively locate the target [14, 26], leading to misleading guidance instead of facilitating tracking.

## C.3 Comparison with All-in-one and OVLM

As emphasized in Sec. 4.2, MemVLT is compared with existing state-of-the-art VLTs that share similar task settings and training dataset configurations to ensure a fair comparison. As shown in Tab. 1 and Tab. A1, the VLTs we compare against are primarily trained using the LaSOT [27], TNL2K [26], RefCOCOg [49], and OTB99-Lang [1] datasets. In addition, two recent VLTs, namely OVLM [72] and All-in-one

Table A2: Comparison with the OVLM.

| Model | TNL2K | | LaSOT | | Average | |
|---|---|---|---|---|---|---|
| | AUC | P | AUC | P | AUC | P |
| OVLM | 64.7 | 69.3 | 67.7 | 74.2 | 66.4 | 72.1 |
| MemVLT | 63.3 | 67.4 | 72.9 | 80.5 | 68.8 | 74.9 |

[73], utilize training dataset configurations that differ significantly from ours. Therefore, we conduct a separate comparison with these two VLTs.

**OVLM.** The comparison results are shown in Tab. A2. Our MemVLT significantly outperforms OVLM on LaSOT (AUC +5.2%) but slightly falls short on TNL2K (AUC -1.4%). Overall, our model demonstrates superior average performance (AUC +2.4%). Besides, although OVLM involves memory mechanism modeling, there are key differences between our method:

1. **Memory Representation**: OVLM assumes the text prompt is a precise long-term cue, using vision and text features to represent short and long-term memories, respectively. However, it overlooks the misalignment between text information and video targets (see Fig. 1 (a)), which can mislead the tracker [14]. Our method addresses this by using both text and vision features to comprehensively represent short and long-term memories, and modulate them with dynamic information.

2. **Memory Interaction**: OVLM focuses solely on the unidirectional modulation of short-term memory features by textual long-term memory, whereas our method achieves bidirectional modulation between long-term and short-term memories. The importance of bidirectional interaction has been highlighted by recent research [23].

In summary, our model provides a more comprehensive representation and interaction of memory information, resulting in superior performance.

**All-in-one.** As shown in Tab. A3, our tracker significantly outperforms All-in-one on all benchmarks except LaSOT$_{ext}$. Overall, the average performance of our model far exceeds that of All-in-one (AUC +4.5%). Although All-in-one uses a larger training dataset to align text and vision modalities, it neglects the modeling

Table A3: Comparison with the All-in-one.

| Model | TNL2K | | LaSOT | | LaSOT$_{ext}$ | | Average | |
|---|---|---|---|---|---|---|---|---|
| | AUC | P | AUC | P | AUC | P | AUC | P |
| All-in-one | 55.3 | 57.2 | 67.3 | 78.5 | 54.5 | 66.0 | 60.4 | 68.6 |
| MemVLT | 63.3 | 67.4 | 72.9 | 80.5 | 52.1 | 59.8 | 64.9 | 71.4 |

of dynamic temporal information. In contrast, MemVLT effectively models temporal information through the memory mechanism, resulting in superior tracking performance.

## C.4 Effect of the Confidence Prediction Module

To predict the confidence score $p_c^t$, we introduce a confidence prediction module and supervise it using the $L_2$ loss between $p_c^t$ and the IoU value. As shown in Tab. A4, we analyze the impact of this module on the model's performance using the TNL2K dataset. Without confidence scores, we lack a criterion for short-term memory selection. Therefore, we conduct tests using the *sliding*

Table A4: Study on the Effect of the Confidence Prediction Module. Results are tested on TNL2K. The best results are highlighted in red.

| # | Confidence Prediction | Storage Method | AUC | $P_{Norm}$ | P |
|---|---|---|---|---|---|
| 1 | | *sliding window* | 62.3 | 79.7 | 66.4 |
| 2 | ✓ | *sliding window* | 62.3 | 79.8 | 66.6 |
| 3 | ✓ | *section-top* | 63.3 | 80.9 | 67.4 |

Table A5: Study on the length of visual queries. Results are tested on TNL2K.

| Length | AUC | $P_{Norm}$ | P |
|---|---|---|---|
| 1 | 62.78 | 79.94 | 66.79 |
| 2 | 62.79 | 80.09 | 66.93 |
| 4 | 62.77 | 80.16 | 66.96 |

Table A6: Study on the length of textual queries. Results are tested on TNL2K.

| Length | AUC | $P_{Norm}$ | P |
|---|---|---|---|
| 1 | 62.78 | 79.94 | 66.79 |
| 2 | 62.75 | 80.67 | 66.92 |
| 4 | 62.62 | 80.54 | 66.73 |

*window* storage method (Tab. A4 (#1)). When the confidence prediction module is introduced but the *sliding window* method is still used (Tab. A4 (#2)), we observe no significant change in performance. However, when the *section-top* storage method is employed, we see a significant improvement in model performance (Tab. A4 (#3)). This demonstrates that our proposed long-term memory storage method effectively leverages the confidence prediction information.

## C.5 The Impact of Query Length

In the process of generating short-term memories, we draw inspiration from the widely acknowledged paradigm of prompt learning [35], which introduces learnable query tokens to understand and represent corresponding features. Specifically, in the generation of visual and textual short-term memories $m_v^t$ and $m_l^t$, we incorporate corresponding query tokens $q_v$ and $q_l$, respectively.

As shown in Tab. A5 and Tab. A6, we conduct the ablation analysis on the lengths of these query tokens. Under the setting of a long-term memory length of 4, we test the influence of different lengths of visual and textual queries on the model performance. The results indicate that there is no substantial variation in model performance with changes in query length. This underscores the adequacy of a query length of 1. Consequently, we default to a query length of 1 in the experiments presented in the paper.

## C.6 Ablation Study on the Backbone and Image Resolution

In Sec. 4, all our experiments are conducted using the HiViT-Base backbone and a $384 \times 384$ image resolution. To provide a more comprehensive understanding of our model's performance, we perform ablation studies by evaluating the model under the ViT-Base backbone and a $256 \times 256$ resolution.

In Tab. A7, we not only evaluate the model's performance using two different backbones but also analyze the impact of our proposed memory mechanism. Comparing #1 and #2, it is evident that the introduction of our memory mechanism improves the AUC by 4.3%. Comparing #1 and #3, we observe that the HiViT backbone provides an AUC improvement of 1.1%, which is consistent with previous research findings [40, 39]. Furthermore, the performance boost brought by our memory mechanism surpasses that of the HiViT-based backbone, indicating that the superior performance of our work primarily originates from the memory mechanism rather than the HiViT-based backbone.

In Tab. A8, we evaluate the performance of our model under two different search image resolutions. As expected, reducing the image resolution slightly degrades performance. However, MemVLT-256 still significantly outperforms several recent models with larger resolutions, such as MMTrack [16] ($384 \times 384$) and QueryNLT [54] ($320 \times 320$) (see Tab. 4.2). This demonstrates that our method maintains strong performance even at lower resolutions.

## C.7 Ablation Study on the Number of SMG Layers

Table A7: Ablation study on the backbone and our memory mechanism. Results are tested on TNL2K.

| # | Backbone | Memory | AUC | $P_{Norm}$ | P |
|---|----------|--------|-----|------------|---|
| 1 | HiViT | ✓ | 63.3 | 80.9 | 67.4 |
| 2 | HiViT | | 59.0 | 77.6 | 62.4 |
| 3 | ViT | ✓ | 62.2 | 79.8 | 66.4 |

Table A8: Ablation study on the image resolution. Results are tested on TNL2K.

| # | Setting | AUC | $P_{Norm}$ | P |
|---|---------|-----|------------|---|
| 1 | MemVLT-256 | 63.3 | 80.9 | 67.4 |
| 2 | MemVLT-384 | 59.0 | 77.6 | 62.4 |

In our proposed MemVLT, the SMG module, responsible for generating short-term memory, plays a critical role. To investigate its impact, we conduct ablation experiments on the number of SMG layers. Specifically, we test the performance of SMG with 1 and 5 layers, and compare the results with the existing model using 3 layers.

As shown in Tab. A9, it is observed that using too few layers (comparing #1 and #2) significantly harms the model's performance, while increasing the number of layers (comparing #2 and #3) further does not provide noticeable performance gains. Therefore, our model effectively demonstrates performance convergence.

Table A9: Ablation study on the number of SMG layers. Results are tested on TNL2K.

| # | Layers | AUC | $P_{Norm}$ | P |
|---|--------|-----|------------|---|
| 1 | 1 | 60.9 | 78.3 | 64.8 |
| 2 | 3 | 63.3 | 80.9 | 67.4 |
| 3 | 5 | 63.2 | 81.0 | 67.5 |

# D   Broader Impact

In this paper, we introduce a novel vision-language tracker named MemVLT. Inspired by the Complementary Learning Systems (CLS) theory, this model emulates human memory systems to adapt given static prompts to dynamically changing targets, thereby achieving effective tracking. Generic object tracking is one of the fundamental problems in computer vision with numerous applications such as video surveillance, robotics, and autonomous vehicles. The vision-language tracking task extends this field by incorporating text modality. Besides providing a new type of human-machine interaction, this task leverages the complementary advantages of visual and language modalities to achieve more promising tracking results. Our research can improve tracking performance while maintaining a reasonable running speed (32 FPS). However, it is of particular concern that this tracker could be misused for illegal surveillance and positioning. Despite achieving promising results, applying this technology to real-world scenarios, such as autonomous driving, remains challenging. To mitigate the risks associated with using MemVLT, we encourage researchers to thoroughly understand the implications of using trackers in specific real-world scenarios.

