# OpenReview forum: "MemVLT: Vision-Language Tracking with Adaptive Memory-based Prompts"
_NeurIPS.cc/2024/Conference — NeurIPS 2024 poster_

### Official Review · Reviewer_jufM · 2024-06-21

**Soundness:** 3
**Presentation:** 3
**Contribution:** 3
**Rating:** 7
**Confidence:** 4

**Summary:**

This paper introduces a memory-based vision-language tracker, called MemVLT. Specifically, the memory storage and memory interaction modules are designed to storage and flexible interaction between short-term and long-term memories, achieving vision-language tracking. A large number of experiments demonstrate the effectiveness of the proposed method.

**Strengths:**

1.the paper is well written and can be easily understood by the reader.
2.The proposed tracker can achieve SOTA performance and adequate ablation experiments are accomplished.

**Weaknesses:**

- The idea of designing a long and short-term memory network based on transformers is not novel enough. Its main technical solution consists of multiple transformer layers and memory strategies, which have been fully explored by many previous approaches, e.g., AQATrack [32].

- Figure 1(a) shows the consistency curves for search frames and linguistic descriptions (and template), respectively. However, there is a lack of improved consistency curves, which helps to demonstrate the effectiveness of the proposed method. In addition, consistency analysis why ResNet is used as the feature extraction network instead of ViT or HiViT.

- Since HiViT is used as the visual encoder, I suspect that the performance gain of MemVLT comes from the visual backbone and not from the proposed long and short-term memory module. I would like the authors to provide comparison experiments under different trackers, e.g., ostrack based on ViT backbone [35], to demonstrate the generalization of the proposed method.

- The authors are requested to provide ablation experiments on the number of SMG and transformer decoder layers to observe the effect on the proposed tracker.

**Questions:**

- What do MIM and MSM stand for in line 146? The paper doesn't specify.

**Limitations:**

The limitations of the method have been accounted for in this paper.

---

> ### Author Rebuttal · Authors · 2024-08-05
>
> **Dear Reviewer jufM,**
>
> Thanks for thoroughly reviewing our work. We appreciate your acknowledgment of our model's strong performance and the quality of our writing. Below, we address your concerns and provide detailed responses:
>
> ---
>
> ### **Weakness 1: novelty of our work**
>
> (1) **Novelty in long and short-term memory modeling:**
> To highlight the differences between our method and existing work based on long and short-term memory, we have provided direct comparisons in **Sections 2.1** and **2.2**. Additionally, please refer to our summary of distinctions in **global response 5**.
>
> (2) **Contributions of our technical solution:**
>
> 1). Considering the strong feature representation capability of transformers, we used vanilla transformer layers [1] and transformer variants with multiple cross-attention layers [2][3] to achieve feature interaction among different elements. This is a commonly adopted approach in mainstream model design. However, it is important to emphasize that our primary contribution lies in the modeling of memory mechanisms using these basic network layers, rather than the design of these transformer layers.
>
> 2). Our CLS-based memory strategy shares similarities with the temporal modeling approaches in some contemporary visual trackers (e.g., AQATrack [1]) but also exhibits distinct differences, which we have directly compared in **Section 2.2**. For instance, in terms of memory storage, our proposed section-top storage strategy significantly outperforms the sliding windows approach used in AQATrack (see **Table 4**).
>
> ---
>
> ### **Weakness 2: regarding Figure 1(a)**
>
> (1) The purpose of the consistency curve in **Figure 1(a)** is to illustrate an inherent fact: static initial prompts (i.e., the template patch and language description) and dynamic targets are difficult to maintain consistently over time. Even without this curve, we can qualitatively observe this fact from **Figure 1(a)**. The curve is a supplementary explanation, and this fact cannot be changed. Therefore, we cannot improve the consistency through model design. This challenge motivates our model design to introduce implicit dynamic memory to better guide tracking. Since these features cannot be directly measured, we cannot demonstrate our method's effectiveness by improving this consistency curve.
>
> (2) As described in **Section B.1**, plotting the consistency curve only requires a pretrained backbone capable of extracting image features, such as ResNet or ViT. It is important to note that the backbone used here differs from the backbone of the tracker, as their purposes differ. To address your concerns, we have plotted the consistency curve based on ViT in the supplementary PDF, and you can see that the curves based on ResNet and ViT are consistent.
>
> ---
>
> ### **Weakness 3: ablation of backbone**
>
> We understand your concerns because previous research [3][4] has demonstrated that the HiViT-based backbone offers performance improvements over the ViT-based backbone, which is why we chose it. To address your concerns, we have conducted the relevant experiments and performed the following analysis:
>
> | # | Backbone | Our Memory Mechanism | AUC  | P_N  | P    |
> |---|----------|----------------------|------|------|------|
> | 1 | HiViT    | √                    | 63.3 | 80.9 | 67.4 |
> | 2 | HiViT    | ×                   | 59.0 | 77.6 | 62.4 |
> | 3 | ViT      | √                    | 62.2 | 79.8 | 66.4 |
>
> (1) We find that our proposed memory mechanism brings a 4.3% AUC improvement (comparing #1 and #2), while the HiViT backbone offers AUC gains of 1.1% (comparing #1 and #3). This indicates that the performance improvement brought by our memory mechanism exceeds that of the HiViT-based backbone.
>
> (2) The experimental results in #3 show that with a ViT-based backbone, our model still performs well, surpassing some of the latest VLTs, such as MMTrack [5] and QueryNLT [6] (see **Table 1**). This demonstrates the generalizability of our method.
>
> ---
>
> ### **Weakness 4: ablation of block layers**
>
> Thanks for your suggestion. We have conducted ablation experiments on the number of block layers. Due to limited time, we have temporarily supplemented the performance of SMG and transformer decoder with 1 and 5 layers and compared them with the existing model at 3 layers. It can be observed that too few block layers (comparing #1 and #2) significantly harms the model's performance, while continuously increasing the block layers (comparing #2 and #3) does not yield noticeable performance gains. Our provided model effectively reflects the convergence performance.
>
> | # | Layers | AUC  | P_N  | P    |
> |---|--------|------|------|------|
> | 1 | 1      | 60.9 | 78.3 | 64.8 |
> | 2 | 3      | 63.3 | 80.9 | 67.4 |
> | 3 | 5      | 63.2 | 81.0 | 67.5 |
>
> ---
>
> ### **Question 1: meaning of symbols**
>
> We apologize for any confusion caused by the abbreviations. The meanings of **MIM (Memory Interaction Module)** and **MSM (Memory Storage Module)** are provided in the overview of the model (**lines 117-127**). We will make these abbreviations bold in the future to enhance the readability of the paper.
>
> ---
>
> We hope this detailed response addresses your concerns. We would appreciate it if you could reconsider your rating. If you have additional feedback or require further clarification, please let us know.
>
> ---
>
> [1] Attention is All You Need, Vaswani et al., in NeurIPS 2017.
>
> [2] Grounding DINO: Marrying DINO with Grounded Pre-Training for Open-Set Object Detection, et al., in arXiv 2023.
>
> [3] Autoregressive Queries for Adaptive Tracking with Spatio-Temporal Transformers, Xie et al., in CVPR 2024.
>
> [4] Explicit Visual Prompts for Visual Object Tracking, Shi et al., in AAAI 2024.
>
> [5] Towards Unified Token Learning for Vision-Language Tracking, Zheng et al., in TCSVT 2023.
>
> [6] Context-Aware Integration of Language and Visual References for Natural Language Tracking, Shao et al., in CVPR 2024.

---

> ### Comment · Reviewer_jufM · 2024-08-12
> **Comment**
>
> Thank you for your response, which addresses all of my concerns. I decided to raise my rating to accept.

---

> ### Author Response · Authors · 2024-08-12
> **Acknowledgment of Rating Increase and Planned Revisions**
>
> **Dear Reviewer jufM,**
>
> We are thrilled to receive your response and deeply honored that our reply has resolved all of your concerns. Your decision to raise the rating is incredibly encouraging for us.
>
> We will incorporate your valuable suggestions and include the aforementioned ablation analyses in the revised manuscript.
>
> **Sincerely wishing you all the best ~~~**

---

### Official Review · Reviewer_unPL · 2024-06-29

**Soundness:** 3
**Presentation:** 2
**Contribution:** 3
**Rating:** 5
**Confidence:** 5

**Summary:**

This paper presents a new visual-language tracking method, called MemVLT, which incorporates memory modeling to adjust static prompts.The design of MemVLT is inspired by the theory of Complementary Learning Systems (CLS), which adapts to changes in the target by mimicking the storage and interaction mechanisms of the human memory system.

**Strengths:**

1. MemVLT can better utilize the memory information for vision-language tracking.
2. MemVLT achieves promising results in several benchmarks.

**Weaknesses:**

1. Performance comparison is unfair. MemVLT is built on HiViT-base and the input resolution is 384 instead of 256. HiViT-base can improve the tracking accuracy compared to ViT, which has been proved in previous researches. However, authors did not provide the ablation study on backbone. The input resolution is 384 larger than previous models such as DecoupleTNL, and larger resolution can bring obvious performance improvement. Ablation study on the input resolution is lacked. Author should clarify the input resolution in Table 1 and add more ablation study on resolution and backbone to prove the improvement comes from the memory mechanism instead of stronger backbone and larger resolution.
2. I do not think the memory mechanism can only be applied for visual tracking without language. In another word, the memory mechanism is also suitable for visual tracking. Authors should add experiments to explore the performance of the memory mechanism in visual tracking.
3. Writing is confusing. The explanation of method is confusing. Authors should try to clarify this part to make it easy-to-follow.

**Questions:**

See Weakness

**Limitations:**

Experiments are not sufficient.

---

> ### Author Rebuttal · Authors · 2024-08-05
>
> **Dear Reviewer unPL,**
>
> Thanks for your careful review of our work. Your acknowledgment of our memory mechanism modeling and its promising performance is invaluable to us. We notice that you are particularly concerned about the model's ablation analysis. Therefore, we have conducted the corresponding experiments and hope that the following experimental analysis will address your concerns.
>
> ---
>
> ### **Weakness 1: more ablation analysis**
>
> （1）**Ablation on backbone:**
>
> As you mentioned, previous research [1][2] has demonstrated that the HiViT-based backbone offers performance improvements over the ViT-based backbone, which is why we chose it. It is important to note that the primary contribution of our paper is the modeling of memory mechanisms rather than backbone design. Therefore, our ablation experiments focus on the former. To address your concerns, we have conducted the relevant experiments, with results shown in the table below:
>
> | # | Backbone | Our Memory Mechanism | AUC  | P_N  | P    |
> |---|----------|----------------------|------|------|------|
> | 1 | HiViT    | √                    | 63.3 | 80.9 | 67.4 |
> | 2 | HiViT    | ×                   | 59.0 | 77.6 | 62.4 |
> | 3 | ViT      | √                    | 62.2 | 79.8 | 66.4 |
>
> Comparing #1 and #2, it is evident that the introduction of our memory mechanism improves AUC by 4.3%. Comparing #1 and #3, we can see that the HiViT backbone offers AUC gains by 1.1%, consistent with previous research conclusions [1][2]. Moreover, the performance improvement brought by our memory mechanism exceeds that of the HiViT-based backbone, demonstrating that the superior performance of our work primarily stems from the memory mechanism rather than the HiViT-based backbone.
>
> （2）**Ablation on image resolution:**
> To our knowledge, vision-language trackers typically report results at a single resolution setting, as seen in DecoupleTNL [3] (256), QueryNLT [5] (320), and MMTrack [6] (384). This limits our comparison to the results provided by DecoupleTNL at a resolution of 256 and is also the reason why we did not perform resolution-related ablation experiments. We will clarify the input resolution in Table 1. To address your concerns, we supplement our experiments at the 256 resolution setting (tested on TNL2K):
>
> | # | Setting    | AUC  | P_N  | P    |
> |---|------------|------|------|------|
> | 1 | MemVLT-256 | 62.1 | 79.6 | 65.9 |
> | 2 | MemVLT-384 | 63.3 | 80.9 |67.4 |
>
> As expected, reducing the image resolution somewhat degrades performance. However, MemVLT-256 still significantly outperforms some recent models with larger resolutions, such as MMTrack [6] (384) and QueryNLT [5] (320) (see **Table 1**). This demonstrates that our method maintains strong performance even at lower resolutions.
>
> ---
>
> ### **Weakness 2: generalization of the model**
>
> Thanks for your suggestion. We have conducted experiments under the vision-only tracking setting. The results demonstrate that our method can generalize to the vision-only tracking task. Specifically, we remove the text-related components from MemVLT, seamlessly transforming it into a vision tracker. As shown in the table below, our proposed Memory Interaction Module (MIM) and Memory Storage Module (MSM) incrementally improve the performance of the vision-only tracker, demonstrating the good generalizability of our method.
>
> | Setting  | AUC  | P_N  | P    |
> |----------|------|------|------|
> | Naive    | 57.4 | 74.9 | 60.6 |
> | +MIM     | 58.3 | 76.1 | 61.0 |
> | +MSM     | 59.6 | 76.8 | 62.3 |
>
> ---
>
> ### **Weakness 3: method description**
>
> We apologize for any confusion caused by the description of our method. Due to the need for temporal modeling, the connections between various modules are relatively complex. For instance, when introducing the memory interaction module, we need to explain the meaning of the memory features used in the memory storage module. We suspect this might have caused the confusion. In the revised version, we will meticulously review our paper's wording and enhance the introduction of key concepts in method section. We hope this will provide readers with a reading experience similar to that of Reviewers BESZ and jufM, who highly commended the readability of our paper.
>
> ---
>
> We believe this detailed response has clarified your concerns. We would appreciate it if you could reconsider your rating based on this additional information. Please reach out if you have further questions.
>
> ---
>
> [1] Autoregressive Queries for Adaptive Tracking with Spatio-Temporal Transformers, Xie et al., in CVPR 2024.
>
> [2] Explicit Visual Prompts for Visual Object Tracking, Shi et al., in AAAI 2024.
>
> [3] Tracking by Natural Language Specification with Long Short-Term Context Decoupling, Ma et al., in ICCV 2023.
>
> [4] Joint Visual Grounding and Tracking with Natural Language Specification, Zhou et al., in CVPR 2023.
>
> [5] Context-Aware Integration of Language and Visual References for Natural Language Tracking, Shao et al., in CVPR 2024.
>
> [6] Towards Unified Token Learning for Vision-Language Tracking, Zheng et al., in TCSVT 2023.

---

> > ### Comment · Reviewer_unPL · 2024-08-08
> > **Details on Ablation Study**
> >
> > Thanks for your reply, which has solved part of my concerns. I am still confused about the ablation study. Firstly, what's the performance of MemVLT when evaluated on TNL2K and LaSOT with ViT backbone and 256 resolution? Secondly, in the response to Weakness 2: generalization of the model, on which dataset did you conduct your experiments and what's the backbone and resolution?

---

> > > ### Author Response · Authors · 2024-08-09
> > > **Further Response on the Details of the Ablation Study**
> > >
> > > **Dear Reviewer unPL,**
> > >
> > > Thanks for your timely response. We are very glad that our previous reply addressed some of your concerns. Regarding the new questions, we have conducted relevant tests and analyses as quickly as possible, and we are eager to share and discuss these new results with you.
> > >
> > > ---
> > >
> > > ### **Question 1: Performance of MemVLT with ViT Backbone and 256 Resolution on TNL2K and LaSOT**
> > >
> > > The three tables in our previous reply were tested on TNL2K, a recent dataset specifically tailored for the vision-language tracking task. To address your concerns, we have tested the performance of MemVLT on TNL2K and LaSOT with different backbone and resolution settings. The comprehensive results are summarized in the table below:
> > >
> > > | Backbone | Resolution | TNL2K AUC | TNL2K P | LaSOT AUC | LaSOT P |
> > > |----------|------------|-----------|---------|-----------|---------|
> > > | ViT      | 256        | 61.2      | 65.0    | 70.5        | 77.3    |
> > > | ViT      | 384        | 62.2      | 66.4    | 71.9      | 79.7    |
> > > | HiViT    | 256        | 62.1      | 65.9    | 71.7      | 79.4    |
> > > | HiViT    | 384        | 63.3      | 67.4    | 72.9      | 80.5    |
> > >
> > > ---
> > >
> > > ### **Question 2: Implementation Details of the Memory Mechanism Generalization Experiment**
> > >
> > > In this experiment, we use the HiViT-based backbone with a resolution of 384 to construct different visual tracker variants and test the results on TNL2K.
> > >
> > > It is important to note that although we conduct the experiment with only this backbone and resolution setting, it sufficiently provides evidence of the generalization capability of our memory mechanism. This is because our memory mechanisms (i.e., memory interaction and memory storage modules) are decoupled from the backbone (see **Sec.3** and **Fig.2**). According to the basic principle of controlled variables, to verify the generalization of our memory mechanism, only the memory mechanism modules need to be adjusted while keeping the backbone and image resolution fixed.
> > >
> > > ---
> > >
> > > We believe that the above responses address the two new questions you raised, and we sincerely hope that you can reconsider your rating in light of these new experimental results. If you have any further questions, please feel free to let us know.

---

> > > > ### Comment · Reviewer_unPL · 2024-08-09
> > > > **Concerns about performance improvement**
> > > >
> > > > Thanks for your response. Based on your experiment results and rebuttal, it seems that the main reason of performance improvement comes from stronger backbone and larger resolution. Firstly, for visual tracking with language, MemVLT with 256 and ViT achieves 61.2 AUC on TNL2K and 70.5 AUC on LaSOT. MemVLT  with 256 and ViT outperforms previous models on TNL2K, while achieves the similar performance on LaSOT with DecoupleTNL. Secondly, the performance of MemVLT with HiViT and 384 resolution on visual object tracking task is the same as ARTrack384 and lower than some other models, which demonstrate that the memory mechanism may not be as effective as the authors claim.
> > > > I'm sorry that I tend to maintain my rating because of the unfair comparison and unclear reason of accuracy improvement.

---

> ### Author Response · Authors · 2024-08-09
> **Further Response on Reviewer's Concerns**
>
> **Dear Reviewer unPL,**
>
> Thanks for your further response.
>
> Regarding your comments on "unfair comparison" and "unclear reason for accuracy improvement," we respectfully disagree. First, we highly value your feedback, which is why we promptly conducted additional model training and testing analyses as soon as we saw the concerns you raised. Second, based on the issues you mentioned, we recognize that you have a deep understanding and professional background in tracking tasks.  However, we believe there may be some misunderstanding regarding our experimental settings and motivations, so we feel it is necessary to provide further clarification to address your concerns about the performance.
>
> ### **Question 1: Is the performance comparison between our model and others truly unfair?**
>
> **1. Comparison with DecoupleTNL**
>
> We have summarized the performance of the DecoupleTNL and our model under different backbone and resolution settings in the table below. The conclusions are as follows:
>
> (1) As you mentioned, previous research has demonstrated that a ViT-based backbone and a 256 resolution negatively impact tracker performance. Despite this, even under these challenging settings (#1), our model still achieves strong performance. Compared to DecoupleTNL, we observe a slight performance decrease on LaSOT (AUC -0.7%), but a significant performance improvement on TNL2K (AUC +4.5%). Considering the number of search frames across the two datasets, our model shows a higher average performance (AUC +1.5%).
>
> (2) From a model perspective, the DecoupleTNL includes both a frame encoder and a video encoder as part of its visual backbone, whereas our model only uses a single image encoder. This means our model is at a disadvantage in terms of input information (i.e., no multi-frame video data input). Despite this, we chose to compare under the same resolution (256) in #4 to ensure as much fairness as possible. Clearly, the performance in #4 significantly exceeds that of #1.
>
> | #  | Model               | TNL2K AUC | TNL2K P | LaSOT AUC | LaSOT P | Average AUC | Average P |
> |----|---------------------|-----------|---------|-----------|---------|-------------|-----------|
> | 1  | DecoupleTNL (256)   | 56.7      | 56.0    | 71.2      | 75.3    | 65.0        | 67.0      |
> | 2  | MemVLT (ViT-256)    | 61.2      | 65.0    | 70.5      | 77.3    | 66.5        | 72.0      |
> | 3  | MemVLT (ViT-384)    | 62.2      | 66.4    | 71.9      | 79.7    | 67.7        | 74.0      |
> | 4  | MemVLT (HiViT-256)  | 62.1      | 65.9    | 71.7      | 79.4    | 67.6        | 73.6      |
> | 5  | MemVLT (HiViT-384)  | 63.3      | 67.4    | 72.9      | 80.5    | 68.8        | 74.9      |
>
> **2. What settings represent a model's final performance to demonstrate SOTA performance?**
>
> To compare models and demonstrate SOTA performance, it is essential to focus on the model's absolute performance. To our knowledge, almost all VLTs consider only one setting for absolute performance when comparing with other models to claim SOTA results. Given that it is common for visual trackers to show results under "additional settings", we have also provided such analyses.
>
> However, it is important to note that when comparing with other models, these "additional settings" are typically limited to resolution. For example, in OSTrack and AQATrack, models' performances at 256 and 384 resolutions are presented to compare with others.
>
> Therefore, the absolute performance of our model can be represented by the results in the table above (#4, #5). Clearly, our model outperforms existing VLTs significantly (see Tab.1), which sufficiently demonstrates that our model achieves SOTA performance.
>
>
> ### **Question 2: What contributes to our strong performance?**
> The explanation for this question will be provided in the next comments.

---

> > ### Author Response · Authors · 2024-08-09
> > **Further Response on Reviewer's Concerns (continue)**
> >
> > ### **Question 2: What contributes to our strong performance?**
> >
> >  **1.Our efforts to analyze what contributes to the performance improvement**
> >
> > (1) Regarding our core contribution, i.e., memory mechanism modeling, we have conducted extensive ablation studies in the paper (see Sec.4.3, Tab.3–5) to explain the impact of different module designs on model performance. The main experimental conclusions can be summarized as follows:
> >
> >    - Tab.3 shows that applying our memory mechanism modeling to both visual and textual cues results in significant performance improvements. This demonstrates that our memory mechanism can enhance the performance.
> >    - Tab.4 explores the effectiveness of our memory storage method within our memory mechanism modeling. The superior performance of our storage method compared to others proves its effectiveness.
> >    - Tab.5 investigates the effectiveness of our memory interaction method within the memory mechanism modeling. By ablating key computational operations, we explain the reasons for the performance improvements brought by our memory interaction method.
> >
> > (2) We noticed that you are particularly concerned about the impact of the encoder backbone and image resolution on the model's performance, as well as the generalization of the memory mechanism to visual tracking tasks. Therefore, we supplemented the relevant ablation experiments. The main conclusions are as follows:
> >
> >    - In the first table of our initial response, we have demonstrated that the strong performance of our model primarily stems from our memory mechanism design, rather than the choice of the HiViT-based backbone.
> >    - In the second table of our initial response, our model maintains SOTA performance even at a 256 resolution, indicating that the strong performance of our model is not due to the choice of a 384 resolution.
> >    - in the third table of our initial response, by using the memory-related modules as variables, we have proven that our memory mechanism modeling can generalize to visual tracking tasks.
> >
> > In summary, we have thoroughly explained the reasons for the performance improvements of our model through extensive experiments. The conclusions can be summarized as follows:
> >
> >    - From an overall architecture perspective, the strong performance of our model primarily comes from our memory mechanism modeling (which is also the core contribution of our work), rather than from the HiViT-based backbone or the 384 resolution.
> >    - Focusing on memory mechanism modeling, on one hand, we have validated the effectiveness of our proposed memory interaction module and memory storage method; on the other hand, we have also confirmed that the memory mechanism can generalize to visual tracking tasks.
> >
> > **2. Regarding the generalization experiment of the memory mechanism**
> >
> > (1) First, we need to reiterate the purpose and design of our experiment. To verify whether our memory mechanism modeling can generalize to visual tracking, we first removed the text-related modules from MemVLT, transforming it into a visual tracker. We then conducted the relevant experiments with the memory-related modules as the only controlled variables. The experimental results (as shown in the third table of our initial response) demonstrate that as we introduce the memory modules (MIM, MSM), the performance of the visual tracker improves significantly. This is sufficient to prove that our memory mechanism modeling can generalize to visual tracking.
> >
> > (2) Based on the second question you raised, it seems that you may have misunderstood the purpose of this experiment as proving that our model in the visual tracking setting should outperform SOTA visual trackers, and that surpassing ARTrack is a measure of generalization. We have the following two points of clarification:
> >
> >    - ARTrack and our model have numerous differences, making it impossible to verify the generalization of our memory mechanism based on the principle of controlled variables.
> >    - We conducted the experiment by removing the text-related components from MemVLT. Compared to recent SOTA models specifically designed for visual tracking, our model is at a significant disadvantage in terms of parameters. For example, ARTrack's decoder has 6 layers, whereas our visual memory-related module has only 3 similar decoder layers. Despite this, our model achieved comparable performance to the SOTA visual trackers, e.g., our model slightly underperforms ARTrack-384 (AUC -0.2%,CVPR 2023) but surpasses AQATrack-384 (AUC+0.2%, CVPR2024) and F-BDMTrack-384 (AUC+1.8%, ICCV 2023).
> >    - To further address your concerns, we are currently training the model in a visual tracking setting with more decoder layers. We will share the results with you as soon as they are available.
> >
> > The above content addresses your concerns regarding "unfair comparison" and "unclear reasons for accuracy improvement." We sincerely hope you can provide further guidance on what steps we can take to improve our rating.

---

> > > ### Comment · Reviewer_unPL · 2024-08-09
> > > **Comment**
> > >
> > > Thanks for your reply. Considering your contribution, I will raise my rating to Borderline accept. I think authors should clarify the comparison condition in the later version and add the ablation study on backbone and resolution.

---

> ### Author Response · Authors · 2024-08-10
> **Acknowledgment of Rating Increase and Planned Revisions**
>
> **Dear Reviewer unPL,**
>
> We sincerely appreciate your positive response.  Your decision to raise the rating is of great importance to us.
>
> We highly value your constructive suggestions. In the revised manuscript, we will clarify our comparison conditions and include the ablation experiments related to backbone and resolution.
>
> Wishing you all the best ~~~

---

### Official Review · Reviewer_GRLs · 2024-07-08

**Soundness:** 3
**Presentation:** 4
**Contribution:** 3
**Rating:** 5
**Confidence:** 5

**Summary:**

This paper introduces MemVLT (Memory-based Vision-Language Tracker), an approach to vision-language tracking (VLT) that incorporates memory mechanisms inspired by human cognition. The key innovation is the use of adaptive memory-based prompts to guide tracking, as opposed to relying solely on initial static prompts.
The model consists of two main components:
-A Memory Storage Module (MSM) that efficiently stores and manages long-term memory information.
-A Memory Interaction Module (MIM) that generates adaptive visual and textual prompts by integrating stored memory with current input.
The authors evaluate MemVLT on several VLT benchmarks, demonstrating state-of-the-art performance, particularly on challenging datasets like MGIT and TNL2K.

**Strengths:**

- The paper introduces a paradigm for VLT by incorporating memory mechanisms inspired by cognitive science, specifically the Complementary Learning Systems theory.
- Strong performance: MemVLT achieves state-of-the-art results on multiple benchmarks, showing significant improvements over existing methods.
- Broader impact discussion: The paper includes a thoughtful discussion of potential societal impacts and limitations.

**Weaknesses:**

- The insights of using adaptive prompts for tracking guidance have been introduced by CiteTracker[1]. More analysis and comparison between them should be provided.
- Short-term and long-term memory have been exploited more in the video object segmentation task like Xmem[4]. More analysis and comparison between them should be provided.
- The authors are suggested to conduct more comparisons with the most recently published VL tracker like[2][3].
- While the model achieves good performance, it may be more computationally intensive than simpler approaches due to the memory mechanisms.

[1] Li X, Huang Y, He Z, et al. Citetracker: Correlating image and text for visual tracking[C]//Proceedings of the IEEE/CVF International Conference on Computer Vision. 2023: 9974-9983.
[2] Zhang C, Sun X, Yang Y, et al. All in one: Exploring unified vision-language tracking with multi-modal alignment[C]//Proceedings of the 31st ACM International Conference on Multimedia. 2023: 5552-5561.
[3] Wu Z, Zheng J, Ren X, et al. Single-model and any-modality for video object tracking[C]//Proceedings of the IEEE/CVF Conference on Computer Vision and Pattern Recognition. 2024: 19156-19166.
[4] Cheng H K, Schwing A G. Xmem: Long-term video object segmentation with an atkinson-shiffrin memory model[C]//European Conference on Computer Vision. Cham: Springer Nature Switzerland, 2022: 640-658.

**Questions:**

The main limitation of this paper lies in its limited novelty. The authors should provide a more detailed comparison based on this weakness.

**Limitations:**

no.

---

> ### Author Rebuttal · Authors · 2024-08-05
>
> **Dear Reviewer GRLs,**
>
> We sincerely appreciate your time and effort in reviewing our work. We are grateful for your recognition of our cognitive science-inspired modeling approach, strong performance, and broader impact discussion. We notice that you are particularly concerned about the differences between our model and other works. Therefore, we have conducted a detailed comparative analysis to highlight the novelty of our method.
>
> ---
>
> ### **Weakness 1: Comparison with CiteTracker**
>
> (1) **Differences in adaptive prompts:**
>
> 1). CiteTracker is a vision tracker. Its adaptive prompts refer to discrete and closed-set text cues identified based on visual target information. This is primarily achieved through image attribute recognition.
>
> 2). Our proposed MemVLT is designed for the vision-language tracking task. It considers textual information that is diverse and open-set in language description, making it a more challenging task. Our adaptive prompts refer to initial multimodal prompts implicitly embedded with dynamic interaction information. Unlike CiteTracker, which only models adaptive textual prompts, our approach models both textual and visual adaptive prompts, aiding in better tracking guidance. Additionally, our adaptive prompts are implicitly obtained through prompt learning, which helps mitigate the impact of classification errors caused by explicit image attribute recognition.
>
> (2) **Performance comparison:** As shown in the table below, our MemVLT significantly outperforms CiteTracker, demonstrating the effectiveness of our adaptive prompts mechanism.
>
> | Model      | TNL2K AUC | TNL2K P | LaSOT AUC | LaSOT P | Average AUC | Average P |
> |------------|-----------|---------|-----------|---------|-------------|-----------|
> | CiteTracker| 57.7      | 59.6    | 69.7      | 75.7    | 64.5        | 68.8      |
> | MemVLT     | 63.3      | 67.4    | 72.9      | 80.5    | 68.8        | 74.9      |
>
> ---
>
> ### **Weakness 2: Comparison with Xmem**
>
> (1) **Key differences in memory mechanism modeling:**
>
> 1). Xmem uses the **Atkinson-Shiffrin memory model**, which emphasizes the representation of short-term, working, and long-term memories. A crucial modeling processing of Xmem is the unidirectional information transfer from short-term to long-term memory. In contrast, our memory model is based on the **Complementary Learning Systems (CLS) theory**, which emphasizes the interaction between short-term and long-term memories. The importance of bidirectional interaction has been highlighted in recent studies [1].
>
> 2). For memory storage and updating processes, Xmem adopts a **top-k** updating method based on usage frequency as the importance indicator. In contrast, our proposed **section-top** method not only considers the importance of memory units but also accounts for the temporal representation range(see **Table 4**).
>
> (2) **Different tasks:** Xmem focuses on vision-only video object segmentation, while MemVLT concentrates on vision-language multimodal tracking. Due to the significant differences in input and output, a direct quantitative comparison is not feasible, and existing VLTs have not conducted performance comparisons with it.
>
> ---
>
> ### **Weakness 3: Comparison with recent SOTA trackers**
>
> (1) **All-in-one:** As shown in the table below, our tracker significantly outperforms All-in-one on all benchmarks except LaSOT_ext. Overall, the average performance of our model far exceeds that of All-in-one (AUC +4.5%). Although All-in-one uses a larger training dataset to align text and vision modalities, it neglects the modeling of dynamic temporal information. In contrast, MemVLT effectively models temporal information through the memory mechanism, resulting in superior tracking performance.
>
> | Model       | TNL2K AUC | TNL2K P | LaSOT AUC | LaSOT P | LaSOT_ext AUC | LaSOT_ext P | Average AUC | Average P |
> |-------------|-----------|---------|-----------|---------|---------------|-------------|-------------|-----------|
> | All-in-one  | 55.3      | 57.2    | 67.3      | 78.5    | 54.5          | 66.0        | 60.4        | 68.6      |
> | MemVLT      | 63.3      | 67.4    | 72.9      | 80.5    | 52.1          | 59.8        | 64.9        | 71.4      |
>
> (2) **Untrack:** In fact, Untrack is not a VL tracker. Although it claims to handle tracking in any modality, it does not include our VLT task, making direct comparison infeasible. Additionally, we found a recent tracker with a similar motivation to Untrack that includes the VLT task, called OneTracker [2], and compared its performance with ours (see table below). Evidently, our model significantly outperforms OneTracker across all metrics. We speculate that the primary reason for this is OneTracker's lack of modeling dynamic temporal information.
>
>
> | Model       | TNL2K AUC | TNL2K P | LaSOT AUC | LaSOT P | Average AUC | Average P |
> |-------------|-----------|---------|-----------|---------|-------------|-----------|
> | OneTracker  | 58.0      | 59.1    | 70.5      | 76.5    | 65.1        | 69.0      |
> | MemVLT      | 63.3      | 67.4    | 72.9      | 80.5    | 68.8        | 74.9      |
>
> ---
>
> ### **Weakness 4: computational efficiency of MemVLT**
>
> We understand your concerns regarding the computational efficiency (e.g., fps, parameters). Actually, we have already included comparisons and analyses in our paper (see **Table 2** and **Section 4.2**). Overall, our model is comparable to the latest trackers in terms of parameters and fps, while demonstrating superior performance.
>
> ---
>
> We hope this detailed response addresses your concerns and would be grateful if you could reconsider your rating. Should you have any additional feedback or require further clarification, please do not hesitate to let us know.
>
> ---
>
> [1] Organizing memories for generalization in complementary learning systems, Sun et al., in Nature 2023.
>
> [2] OneTracker: Unifying Visual Object Tracking with Foundation Models and Efficient Tuning, Hong et al., in CVPR 2024.

---

> > ### Comment · Reviewer_GRLs · 2024-08-12
> >
> > Thank you for addressing most of my concerns in your response. I recommend that the authors add the above comparisons to the manuscript. This addition would enhance the paper's clarity. I decide to raise my rating to the borderline accept.

---

> ### Author Response · Authors · 2024-08-12
> **Acknowledgment of Rating Increase and Planned Revisions**
>
> **Dear Reviewer GRLs,**
>
> Thanks for your response. We are honored that our reply has resolved most of your concerns. Your decision to raise the rating is of great significance to us.
>
> We will incorporate your valuable suggestions and include the comparisons mentioned above in the revised manuscript.
>
> **Wishing you all the best !!!**

---

### Official Review · Reviewer_BESZ · 2024-07-11

**Soundness:** 2
**Presentation:** 3
**Contribution:** 3
**Rating:** 6
**Confidence:** 4

**Summary:**

This paper proposes a Memory-based Visual-Language Tracker based on the complementary learning system theory. Memory mechanism modeling facilitates the generation of adaptive prompts to effectively guide the tracking process. Extensive experiments demonstrate that MemVLT achieves new state-of-the-art performance.

**Strengths:**

1) Integrating memory mechanisms with adaptive prompts is interesting for vision-language tracking.
2) This paper is well-writing and well-organized, and includes appropriate citations and references.
3) The experiments across various benchmarks validate MemVLT's performance compared to other trackers.

**Weaknesses:**

1) For experiments, recent SOTA trackers are missing in Table 1, such as MMA (CVPR 2024), CAI (CVPR 2024), OVLM (TMM23) ...
2) OVLM is also a recent vision-language memory network for tracking, but the authors have not dicuss and compare it.

**Questions:**

1) The long and short-term memory have been widely used in visual object tracking, so it is not a new technique. What's the difference for vision-language tracking?
2) Is it general to apply the proposed memory prompts for single-modality visual tracking framework?

**Limitations:**

In addition to performance, tracking speed (fps) is also need to be compared and discussed for real-time applications.

---

> ### Author Rebuttal · Authors · 2024-08-05
>
> **Dear Reviewer BESZ,**
>
> Thanks for taking the time to review our work. We appreciate your recognition of our interesting insights, good writing, and strong performance. We hope the following responses will address your concerns.
>
> ---
>
> ### **Weakness 1 & 2: comparison with recent SOTA trackers**
>
> (1) **MMA [1]:** We found that MMA is not a tracker. It focuses on vision-language pre-training, with its primary downstream task being image classification. Due to the significant gap, a direct comparison is not feasible.
>
> (2) **CAI [2]:** The tracker proposed in the CAI paper is called **QueryNLT**, which we have already compared in our paper (see **Table 1**). Our model outperforms QueryNLT across all benchmarks. For instance, on the TNL2K, our model achieves a 6.4% higher in AUC. QueryNLT modulates multimodal prompts by filtering tokens based on visual-text similarity. In contrast, our modulation approach embeds dynamic information into the multimodal prompts, avoiding the potential loss of important tokens due to similarity calculation errors. We suspect this accounts for the inferior performance of QueryNLT.
>
> (3) **OVLM [3]:** The comparison results are shown in the table below. Our MemVLT significantly outperforms OVLM on LaSOT (AUC +5.2%) but slightly falls short on TNL2K (AUC -1.4%).  Overall, our model demonstrates superior average performance (AUC +2.4%). Although OVLM  involves memory mechanism modeling, there are key differences between our method (addressing weakness 2):
>
> 1). **Memory Representation:** OVLM assumes the text prompt is a precise long-term cue, using vision and text features to represent short and long-term memories, respectively. However, it overlooks the misalignment between text information and video targets (see **Fig 1(a)**), which can mislead the tracker [4]. Our method addresses this by using both text and vision features to comprehensively represent short and long-term memories, and modulate them with dynamic information.
>
> 2). **Memory Interaction:** OVLM focuses solely on the unidirectional modulation of short-term memory features by textual long-term memory, whereas our method achieves bidirectional modulation between long-term and short-term memories. The importance of bidirectional interaction has been highlighted by recent research [5].
>
> In summary, our model provides a more comprehensive representation and interaction of memory information, resulting in superior performance.
>
> | Model  | TNL2K AUC | TNL2K P | LaSOT AUC | LaSOT P | Average AUC | Average P |
> |--------|-----------|---------|-----------|---------|-------------|-----------|
> | OVLM   | 64.7      | 69.3    | 67.7      | 74.2    | 66.4        | 72.1      |
> | MemVLT | 63.3      | 67.4    | 72.9      | 80.5    | 68.8        | 74.9      |
>
> ---
>
> ### **Question 1: novelty in long and short-term memory modeling**
>
> (1) To highlight the differences between our method and existing trackers based on long and short-term memory, we have provided direct comparisons in **Sections 2.1** and **2.2**. Additionally, please refer to our summary of distinctions in **global response 5**.
>
> (2) For the VLT task, introducing textual cues brings new challenges. Due to the modality differences between text and vision cues, previous visual memory modeling methods may not directly apply to textual memory. For example, acquiring the memory unit in TrDiMP[6] relies on RoI processing in the search image, which cannot be applied to the text modality. Our tracker proposes a novel approach to represent textual memory features based on the prompt learning method. The table below also demonstrates the necessity of specifically designing textual memory information.
>
> | Our text memory mechanism | AUC  | P_N  | P    |
> |------------------|------|------|------|
> | ×     | 59.0 | 77.6 | 62.4 |
> | √      | 62.4 | 79.9 | 66.5 |
>
> ---
>
> ### **Question 2: generalizability of our memory mechanism**
>
> As introduced in the **global response 4**, we conduct experiments under a vision-only tracking setting and find that our method can be generalized to the vision tracker. Specifically, we remove the text-related components from MemVLT, seamlessly transforming it into a vision tracker. As shown in the table below, the proposed Memory Interaction Module (MIM) and Memory Storage Module (MSM) incrementally improve the performance of the vision-only tracker. This validates that our method can generalize to a visual tracking framework.
>
> | Setting | AUC  | p_N  | P    |
> |---------|------|------|------|
> | Naive   | 57.4 | 74.9 | 60.6 |
> | +MIM    | 58.3 | 76.1 | 61.0 |
> | +MSM    | 59.6 | 76.8 | 62.3 |
>
> ---
>
> ### **Limitations : efficiency of MemVLT**
>
> We have already included comparisons and analyses of efficiency in our paper (see **Table 2** and **Section 4.2**). Overall, our model is comparable to the latest trackers in terms of parameters and fps, while demonstrating superior performance. Additionally, our model achieves a speed of 32 fps, which meets the requirements for real-time processing.
>
> ---
>
> We hope this detailed response addresses your concerns. We would appreciate it if you could reconsider your rating. If you have additional feedback or require further clarification, please let us know.
>
> ---
>
> [1] MMA: Multi-Modal Adapter for Vision-Language Models, Yang et al., in CVPR 2024.
>
> [2] Context-Aware Integration of Language and Visual References for Natural Language Tracking, Shao et al., in CVPR 2024.
>
> [3] One-Stream Vision-Language Memory Network for Object Tracking, Zhang et al., in TMM 2023.
>
> [4]  Joint visual grounding and tracking with
>  natural language specification, Zhou et al., in CVPR 2023.
>
> [5] Organizing memories for generalization in complementary learning systems, Sun et al., In Nature 2023.
>
> [6] Transformer meets tracker: Exploiting temporal context for robust visual tracking, Wang et al., in CVPR 2021.

---

> > ### Author Response · Authors · 2024-08-12
> > **Further Clarification and Insights Based on Reviewer Discussions**
> >
> > **Dear Reviewer BESZ,**
> >
> > Thank you once again for taking the time to review our paper. Your insightful suggestions have provided us with valuable inspiration.
> >
> > This comment is not intended to rush you, but rather to share with you some discussions we've had with other reviewers who raised similar concerns. We sincerely hope that by sharing these insights, we can further address any remaining concerns you may have.
> >
> > ---
> >
> > ### **Weakness 1 & 2: Comparison with recent SOTA trackers**
> >
> > Regarding the recent models you mentioned, namely MMA (CVPR 2024), CAI (CVPR 2024), and OVLM (TMM23), we conducted direct comparisons and analyses in our initial rebuttal. These analyses highlighted several distinct differences between our model and existing works, while also demonstrating the superior tracking performance of our approach.
> >
> > In addition, **Reviewer GRLs** also brought up recent works such as CiteTracker (ICCV 2023), Xmem (ECCV 2022), All-in-one (ACMMM 2023), Untrack (CVPR 2024), and OneTracker (CVPR 2024). After thorough comparison and analysis, **Reviewer GRLs** acknowledged the novelty and effectiveness of our method and subsequently raised our rating.
> >
> > We greatly appreciate the references to these recent works provided by both you and **Reviewer GRLs**. We will cite them in future versions of our paper to further enhance our work.
> >
> > ---
> >
> > ### **Question 1: Novelty in long and short-term memory modeling**
> > Similar to your concerns, **Reviewer jufM** also focused on the distinctions between our CLS-based long and short-term memory modeling and existing work. In addition to the responses we provided in our initial reply to you, we conducted further analysis on the technical contributions of our approach as suggested by **Reviewer jufM**. After discussing, **Reviewer jufM** confirmed that our response fully addressed his concerns and subsequently raised our rating. We sincerely hope that our response can also resolve your concerns in this question.
> >
> > ---
> >
> > ### **Question 2: Generalizability of our memory mechanism**
> >
> > Your insightful suggestion regarding the generalizability of our memory mechanism was echoed by **Reviewer unPL**. In addition to presenting and analyzing the third table from our initial response to you, we conducted additional analyses on the experimental setup in our discussions with **Reviewer unPL**. After these discussions, **Reviewer unPL** agreed that our memory mechanism generalizes well to visual trackers and subsequently raised our rating. We believe that these responses and discussions should effectively address this issue.
> >
> > ---
> >
> > ### **Limitations: Efficiency of MemVLT**
> >
> > Similarly, we noticed that **Reviewer GRLs** was also concerned about the computational efficiency of our model. After reviewing Table 2 and Section 4.2 of our paper, **Reviewer GRLs**' concerns were alleviated, and our rating was raised.
> >
> > ---
> >
> > We truly appreciate the time and effort you've invested in reviewing our paper. We believe that the additional information provided above will help clarify your concerns, and We sincerely hope you can provide further guidance on what steps we can take to improve our rating.

---

> > > ### Comment · Reviewer_BESZ · 2024-08-13
> > >
> > > Thanks for your enthusiastic response, which addressed most of my concerns. I decide to raise my rating to the borderline accept.

---

> > > > ### Author Response · Authors · 2024-08-13
> > > > **Acknowledgment of Rating Increase and Planned Revisions**
> > > >
> > > > **Dear Reviewer BESZ,**
> > > >
> > > > Thank you very much for your further feedback. We are honored that our response has resolved most of your concerns, and we are delighted that you have raised our rating to "Weak Accept."
> > > >
> > > > We will also incorporate your valuable suggestions and include the results from the discussions above in the revised manuscript.
> > > >
> > > > **Wishing you all the best !**

---

### Author Rebuttal · Authors · 2024-08-05

**Dear Reviewers,**

We would like to thank all the reviewers for their feedback and help in improving our work. We are glad that they acknowledge the **interesting insights** from introducing CLS theory into the VLT task (Reviewers BESZ, GRLs, and unPL), **adequate ablation analysis** (Reviewer jufM), **strong performance** (Reviewers BESZ, GRLs, unPL, and jufM), and **clear writing** (Reviewers BESZ and jufM). Before addressing each reviewer's concerns individually, we provide preliminary responses to some common questions raised:

---

### **1. Model computational efficiency**

Like Reviewers BESZ (Limitations) and GRLs (Weaknesses 4), we also think that computational efficiency (e.g., fps, parameters) is an essential factor. Therefore, we have already analyzed this in our paper (see **Table 2** and **Section 4.2**). The results indicate that our MemVLT is comparable to the latest trackers in terms of parameters and fps, while demonstrating superior performance.

---

### **2. Comparison with recent work**

We thank Reviewers BESZ (Weaknesses 1&2) and GRLs (Weaknesses 1,2&3) for pointing out some recent VL trackers and related models in other tasks. We have provided detailed comparisons and analyses in the individual rebuttals addressed to each reviewer. Our findings are as follows:

(1) For all VL trackers mentioned by the reviewers, our model still shows the **best average performance**, e.g., exceeding OVLM by 2.4%, and All-in-one by 4.5%. Thus, these latest trackers do not alter the conclusion that MemVLT achieves SOTA performance.

(2) We have also thoroughly compared models from other tasks mentioned by the reviewers. Although similar terminologies like "adaptive prompts" are used in CiteTracker, our model design significantly differs from theirs.

---

### **3. Further ablation analysis on backbone**

Since the core contribution of our work revolves around memory interaction and storage modeling, our ablation experiments are focused on these aspects. We are grateful for the suggestions from Reviewers unPL (Weaknesses 1) and jufM (Weaknesses 3) regarding additional ablation analysis on the backbone (HiViT v.s. ViT).   We have conducted corresponding experiments and will present the detailed results in the rebuttals to each reviewer. Key findings include:

(1) The performance improvement from our memory mechanism surpasses that provided by the HiViT-based backbone.

(2) Even without a HiViT-based backbone, our tracker still achieves SOTA performance.

This confirms the efficacy of our core contribution in memory mechanism modeling.

---

### **4. Generalizability of memory mechanism modeling**

We thank Reviewers BESZ (Weaknesses 2) and unPL (Weaknesses 2) for their suggestion to analyze whether our method can generalize to the vision tracker. We conducted relevant experiments, and the results demonstrate that our proposed memory interaction and storage mechanisms progressively enhance the performance of the vision tracker. This indicates that our memory mechanism modeling can indeed generalize to visiual tracking.

---

### **5. Novelty in long and short-term memory modeling**

Reviewers BESZ (Questions 1), GRLs (Weaknesses 2), and jufM (Weaknesses 1) are particularly interested in how our work differs from existing methods based on long and short-term memory modeling. It is important to note that long and short-term memory is a generalized concept that includes specific memory theories such as Atkinson-Shiffrin and CLS. The CLS theory, which inspires our module design, is based on the latest discoveries in cognitive science from the **past two years** [1][2]. To highlight the novelty and distinction of our method, we have directly compared our approach with existing work in **Sections 2.1** and **2.2**. To summarize, the main distinctions are:

 (1) **Modeling motivation:** Existing works on long and short-term memory modeling, such as DecoupleTNL, typically aim to represent long and short-term memory and guide tracking with the initial prompt. However, they overlook scenarios where significant deviations exist between static prompts and dynamic targets (see **Fig.1(a)**), and using the initial prompts constantly might mislead tracking. In contrast, our work is motivated by adjusting the initial prompts using long and short-term memory to generate prompts that adapt to dynamic target changes. We then use these adaptive prompts to guide tracking.

(2) **Memory theory:**  To our knowledge, our work is the first to introduce CLS theory into the VLT task. This theory emphasizes the process of obtaining adaptive memory through long and short-term memory interaction, aligning with our modeling motivation for adaptive prompts.

(3) **Model Design:** Inspired by the CLS theory, we design a memory interaction module (MIM) and a memory storage module (MSM), which significantly differ from existing trackers. For the MIM, we emphasize adjusting static prompts through memory interaction. For the MSM, our proposed section-top storage method takes into account both the importance of memory units and the length of temporal representation, surpassing existing sliding window storage methods.

(4) **Performance:** As all reviews have emphasized, our model achieves strong performance across all benchmarks, surpassing existing VLTs based on long and short-term memory modeling. This validates the soundness of our modeling motivation and the effectiveness of our model design.

---

Below we have responded to every reviewer individually. In the discussion phase, please enlighten us on anything we can do to further improve the paper.

---

[1] Organizing memories for generalization in complementary learning systems, Sun et al., in Nature 2023.

[2] A generative model of memory construction and consolidation, Eleanor et al., in Nature 2024.

---

### Decision · Program_Chairs · 2024-09-25

**Decision:**

Accept (poster)

**Comment:**

The paper initially received mixed borderline reviews: 3 BR, 1 BA. The major concerns raised by the reviewers were:

1) missing SOTA trackers for comparison  (BESZ, GRLs)
2) What's the novelty of vision-language tracking? (BESZ)
3) the generality of the proposed memory prompts? (BESZ)
4) novelty of adaptive prompts for tracking compared to CiteTracker? (GRLs)
5) novelty of short- and long-term memories compared to Xmem, AQATrack (GRLs, jufM)
6) computational efficiency? (GRLs)
7) unfair comparison due to different backbones, missing corresponding ablation study (unPL, jufM)
8) memory mechanism could also be applied to visual tracking (unPL)
9) consistency curves don't improve (jufM)
10) missing ablation study on the number of SMG and layers. (jufM)

The authors wrote a response. After the response, reviewers had a fruitful discussion with the authors and eventually all reviewers raised their ratings to 2BA, 1WA, and 1A. The authors should update the paper according to the reviews and discussions.